# GeneFlow: Translation of Single-cell Gene Expression to Histopathological Images via Rectified Flow

**Mengbo Wang**[*1,4]     **Shourya Verma**[*1]     **Aditya Malusare**[2,4]     **Luopin Wang**[1,4]

**Yiyang Lu**[2]     **Vaneet Aggarwal**[1,2,4]     **Mario Sola**[3,4]     **Ananth Grama**[†1,4]

**Nadia Atallah Lanman**[†3,4]

Purdue University

## Abstract

Spatial transcriptomics technologies can be used to align transcriptomes with histopathological morphology, presenting exciting new opportunities for biomolecular discovery. Using spatial transcriptomic gene expression and corresponding histology data, we construct a novel framework, GeneFlow, to map single- and multi-cell gene expression onto paired cellular images. By combining an attention-based RNA encoder with a conditional UNet guided by rectified flow, we generate high-resolution images with different staining methods (e.g., H&E, DAPI) to highlight various cellular/ tissue structures. Rectified flow with high-order ODE solvers creates a continuous, bijective mapping between expression and image manifolds, addressing the many-to-one relationship inherent in this problem. Our method enables the generation of realistic cellular morphology features and spatially resolved intercellular interactions under genetic or chemical perturbations. This enables minimally invasive disease diagnosis by revealing dysregulated patterns in imaging phenotypes. Our rectified flow based method outperforms diffusion methods and baselines in all experiments. https://github.com/wangmengbo/GeneFlow.

## 1 Introduction

Spatial transcriptomics has revolutionized our understanding of gene expression within tissue architecture, providing unprecedented insights into biological processes and disease mechanisms [1, 2]. Combined with co-registered high-resolution histology images, spatial transcriptomes provide exciting opportunities for understanding the relationship between cellular transcriptomes and the corresponding image phenotypes. Existing computational approaches focus primarily on inferring gene expression from histological images [3, 4]. We address the largely unexplored inverse problem: Generating realistic histopathology images from transcriptomic data.

Spatial transcriptomics technologies such as Slide-seq, Stereo-seq, Visium and Xenium [5, 6, 7, 8] simultaneously capture morphological features through histological imaging (stained with H&E or DAPI ) and transcriptomic profiles through spatially resolved gene expression measurements. This multi-modal approach reveals tissue architecture, cellular heterogeneity, and molecular mechanisms with greater depth than either modality alone, particularly in complex tissues where spatial organization impacts function. Previous machine learning applications in this field have focused on the

---

*Equal Contribution (`wang4887@purdue.edu`, `verma198@purdue.edu`), † Corresponding Authors
[1] Computer Science, [2] Industrial Engineering, [3] Comparative Pathobiology, [4] Institute For Cancer Research.

39th Conference on Neural Information Processing Systems (NeurIPS 2025).

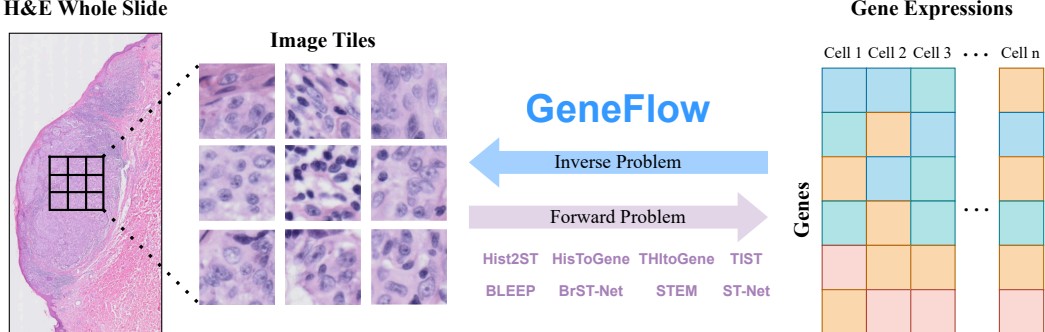

Figure 1: Mapping from histopathology images to gene expression (forward problem) and vice versa (inverse problem) through GeneFlow.

forward problem of predicting gene expression from histology images or integrating both modalities to enhance spatial clustering and gene expression analysis [9, 10, 11, 12, 13, 14, 15, 16]. However, the inverse problem, illustrated in Figure 1, remains largely unexplored. We introduce GeneFlow, the first method that generates histopathology images from single- and multi-cell gene expression transcriptomic profiles.

The ability to generate histopathology images from transcriptomic data has profound implications for cancer research and precision medicine. Cancer exhibits complex molecular alterations that manifest in diverse histological patterns, making it an important use case for our generative approach [17, 18, 19, 20]. Although histopathology remains the gold standard for cancer diagnosis, molecular profiling has become increasingly important for understanding cancer biology and guiding treatment decisions [21, 22, 23]. Studies using spatial transcriptomics data across diverse cancer types highlight the potential impact of integrating both data types. GeneFlow bridges these critical modalities with several potential applications, such as visualizing the histological manifestations of specific gene expression patterns, hypothesis generation, and biomarker discovery.

## 2 Related Work

**Inferring Transcriptomes from Histology Images.** The forward problem of predicting gene expression from histology images has been extensively studied. Wang et al. [9] provided a comprehensive benchmark for spatial transcriptome prediction, revealing variability across tissue types and platforms. Transformer-based models such as HisToGene [10] and THIToGene [11] capture long-range spatial dependencies in H&E images to infer gene expression patterns. TIST [13] introduced a self-supervised framework leveraging unlabeled histology data, while BLEEP [14] employed contrastive learning to align histological and transcriptomic features in a shared latent space, enabling bidirectional cross-modal queries. HIST2ST [15] used graph neural networks to model cellular interactions, and STEM [16] integrated multi-scale tissue representations to capture hierarchical biological organization. These methods collectively demonstrate the feasibility of inferring molecular profiles from morphology but address the forward problem, in contrast to our framework, which tackles the inverse mapping from transcriptomes to histology.

**Mapping Transcriptomes to Histology Images.** While predicting gene expression from histology has been widely explored, the inverse task of generating histopathology images from transcriptomic data remains largely unaddressed. To our knowledge, no prior work directly synthesizes realistic single- or multi-cell H&E or DAPI-stained images from spatial transcriptomics. Existing approaches only partially tackle this problem: RNA-GAN [24] generates histology-like tiles from bulk RNA-seq but lacks single-cell resolution and spatial structure modeling, while HistoXGAN [25] reconstructs cancer histology using multimodal embeddings that depend on pre-extracted histological features rather than gene expression. Consequently, the inverse mapping from transcriptomes to histology, particularly within spatial transcriptomics, remains an open challenge. Our framework, **GeneFlow**, is the first to directly address this task by employing rectified flow to learn the high-dimensional correspondence between spatial gene expression, cellular morphology, and tissue organization, establishing a foundation for bidirectional multi-modal integration in spatial biology.

**Rectified Flow.** Rectified flow is a generative modeling framework that constructs continuous bijective mappings between probability distributions via ordinary differential equations (ODEs). Proposed by Liu et al. [26], it extends normalizing flows [27] and continuous normalizing flows [28] by learning straight-line transport paths for efficient and stable generation. Unlike diffusion models [29] that rely on stochastic Markovian denoising, rectified flow deterministically transports probability mass between distributions. Connections to optimal transport [30] further explain its improved sample quality over diffusion models. In this work, we apply rectified flow to the transcriptomic domain, conditioning the flow on gene expression features through an attention mechanism [31] that modulates trajectory dynamics. High-order ODE solvers are employed for precise integration, capturing the nonlinear correspondence between transcriptomic profiles and histological structures.

## 3 Methods

We formulate the problem of generating histopathological images from gene expression data as follows. Given a single-cell resolution spatial transcriptomics dataset $\mathcal{D} = (X_i, I_i)i = 1^N$, where $X_i \in \mathbb{R}^{C_i \times G}$ represents the gene expression matrix for the $i$-th image tile with $C_i$ cells and $G$ genes, and $I_i \in \mathbb{R}^{H \times W \times K}$ denotes the corresponding histopathological image, our goal is to learn a mapping function $f\theta : \mathbb{R}^{C \times G} \to \mathbb{R}^{H \times W \times K}$ that generates realistic histopathological images from gene expression profiles. Our rectified flow approach constructs a continuous bijective mapping between a simple prior distribution (Gaussian noise) and the target distribution of histopathological images conditioned on gene expression embedding and extra control embeddings such as the number of cells. The model learns a time-dependent vector field $v_\theta(x(t), t, X)$ that guides the transformation from random noise $x(0) \sim \mathcal{N}(0, I)$ to a realistic histopathological image $x(1) \approx I$ conditioned on gene expression matrix $X$. The input to our model consists of single-cell or multi-cell gene expression matrices, where each row corresponds to a gene and each column represents a cell. The output is a high-resolution (256×256 pixels) histopathological image with multiple channels, including H&E staining and optional auxiliary channels such as DAPI for visualizing nuclei.

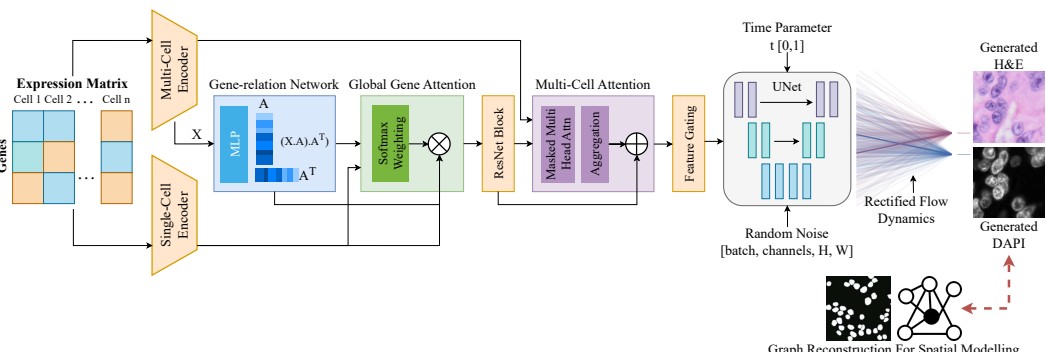

Figure 2: Architecture of the GeneFlow model for mapping transcriptomes to histology images.

### 3.1 GeneFlow Architecture

Our GeneFlow architecture, shown in Figure 2, establishes a framework to translate RNA expression data into histology images using rectified flow dynamics. We develop two distinct encoding pathways to handle different biological contexts: a single-cell encoder and a multi-cell encoder. The single-cell encoder generates image tiles with gene expression of one selected cell while the multi-cell encoder generates image tiles with gene expressions of many cells.

**Gene Relation Network:** Our encoder processes a batch of image patches, where each patch contains multiple cells, represented by a tensor $X \in \mathbb{R}^{B \times C_{\max} \times G}$, with $B$ as batch size, $C_{\max}$ as maximum number of cells per patch, and $G$ as the gene dimension. First, we flatten this tensor for processing $X_{\text{flat}} = \text{reshape}(X) \in \mathbb{R}^{BC_{\max} \times G}$, then the base network produces cell-specific embeddings that are used to predict parameters for a low-rank factorization of gene-gene relationships $E = f_{\text{base}}(X_{\text{flat}}) \in \mathbb{R}^{BC_{\max} \times 256}$. This factorization allows us to model complex interactions between genes while keeping the parameter count manageable $[U, V] = f_{\text{factors}}(E)$. Here, $U \in \mathbb{R}^{BC_{\max} \times G \times K}$

and $V \in \mathbb{R}^{BC_{\max} \times K \times G}$ are cell-specific factor matrices with rank $K$ significantly smaller than the gene dimension $G$. The interaction expression values incorporate learned gene relationships, where $\alpha$ is a scaling factor (tuned during experiments to 0.1) that controls the influence of the learned relationships: $X_{\text{inter}}[i,j] = X[i,j] + \alpha \cdot ((X[i,j] \cdot U[i \cdot C_{\max} + j]) \cdot V[i \cdot C_{\max} + j])$.

**Global Gene Attention:** We apply global gene attention weights to focus on biologically relevant genes, and the weighted expressions are processed through a deep neural network with residual connections to produce cell embeddings with dimension $D$.

$$X_{\text{weight}}[i,j] = X_{\text{inter}}[i,j] \odot \text{softmax}(a) \tag{1}$$

$$H = \text{reshape}(f_{\text{cell\_enc}}(X_{\text{weight,flat}})) \in \mathbb{R}^{B \times C_{\max} \times D} \tag{2}$$

**Multi-Head Cell Attention:** A critical challenge in tissue modeling is aggregating information from variable numbers of cells per patch. We implement a mask to handle this variable-length input $M[i,j] = 1$ if $j < \text{num\_cells}[i]$, otherwise 0. Our multi-head attention mechanism learns to focus on relevant cells, allowing the model to identify cell populations that contribute most significantly to the tissue's visual characteristics:

$$A = f_{\text{cell\_attn}}(H) \in \mathbb{R}^{B \times C_{\max} \times H_{\text{agg}}} \tag{3}$$

$$A_{\text{masked}} = A \cdot M - (1 - M) \cdot \infty \tag{4}$$

$$A_{\text{weights}} = \text{softmax}(A_{\text{masked}}^T) \in \mathbb{R}^{B \times H_{\text{agg}} \times C_{\max}} \tag{5}$$

Each attention head $h$ produces a different weighting over cells, enabling the model to capture various aspects of cellular composition. The attention weights are applied to head-specific projections of cell embeddings:

$$P^{(h)} = f_{\text{proj}}^{(h)}(H) \in \mathbb{R}^{B \times C_{\max} \times D} \tag{6}$$

$$Z_{\text{agg}} = \frac{1}{H_{\text{agg}}} \sum_{h=1}^{H_{\text{agg}}} A_{\text{weights}}[:,h,:] \cdot P^{(h)} \in \mathbb{R}^{B \times D} \tag{7}$$

This design allows various heads to specialize in different aspects of cell behavior, such as identifying rare cell types or focusing on cells with distinctive expression patterns. The outputs from all heads are aggregated and processed through a final encoding layer with feature gating $Z_{\text{gate}} = f_{\text{final}}(Z_{\text{agg}})$, with $Z_{\text{final}} = Z_{\text{gate}} \odot \sigma(W_g \cdot Z_{\text{final}})$. This feature gating mechanism allows the model to selectively emphasize important features in the final representation. This helps control information flow and improves gradient propagation during training.

**UNet Architecture:** Both encoding pathways condition a shared UNet backbone [32] that implements the rectified flow dynamics. The UNet consists of a series of downsampling blocks, a middle block, and upsampling blocks with skip connections. The RNA embedding $z$ (either $z_{\text{single}}$ or $z_{\text{multi}}$) is combined with a time embedding $\gamma(t)$: $\gamma(t) = \text{Embed}(t) \in \mathbb{R}^{4d}$, where $d$ is the base model channel dimension (128 in our implementation). Each residual block in the UNet incorporates these embeddings: $h_{\text{out}} = h_{\text{in}} + \text{Conv}(\text{SiLU}(\text{GroupNorm}(h_{\text{in}}) + \text{Linear}(\gamma(t) + \text{Linear}(z))))$. The UNet predicts the velocity field $v_\theta(x,t)$ that guides the generative process from random noise to fully-formed histological images. During training, the model learns to match the ground truth velocities derived from the rectified flow path: $\mathcal{L}(\theta) = \mathbb{E}x_1, t, \text{noise} \left[ |v\theta(x(t),t) - v^*(x(t),t)|^2 \right] + \lambda |W_1|_1$, where $\lambda = 0.001$ is a regularization parameter and $W_1$ represents the weights of the first layer in the encoder, encouraging sparsity in gene utilization.

The advantages of this method lie in low-rank factorization of gene relationships and the multi-head attention mechanism for cell aggregation. These encoding techniques capture meaningful interactions between gene expressions without requiring extra parameters, making the model more efficient and less prone to overfitting. It also provides explainability by revealing which cells contribute most to the tissue's visual characteristics. The encoder also handles variable numbers of cells per image patch, making it robust for real-world applications where cell density varies significantly across samples.

## 3.2 Generative Modeling With Rectified Flow

Our generative modeling with rectified flow defines a deterministic mapping between noise and data distributions via a continuous-time ODE: $\frac{dx(t)}{dt} = v_\theta(x(t),t), \quad t \in [0,1]$, where $v_\theta$ is a learnable

vector field parameterized by neural network parameters $\theta$. Unlike diffusion models with stochastic trajectories, rectified flow employs straight-line dynamics for efficient and stable generation. For each data point $x_1$, we construct a sinusoidal interpolation path with small stochastic perturbation $\epsilon_t = (1-t)\sigma z$, where $\sigma = 0.05$ and $z \sim \mathcal{N}(0, I)$:

$$x(t) = \sin(t\pi/2)\, x_1 + (1 - \sin(t\pi/2))\, \text{noise} + \epsilon_t, \tag{8}$$

$$v^*(x(t), t) = \frac{dx(t)}{dt} = (x_1 - \text{noise})\frac{\pi}{2}\cos(t\pi/2) - \frac{d\epsilon_t}{dt}, \tag{9}$$

$$\mathcal{L}(\theta) = \mathbb{E}_{x_1, t, \text{noise}}\left[\|v_\theta(x(t), t) - v^*(x(t), t)\|^2\right]. \tag{10}$$

Here, $t \sim \mathcal{U}[0, 1]$. A noise schedule $\sigma(t) = \sigma_{\min} + (\sigma_{\max} - \sigma_{\min})(1-t)$, $\sigma_{\min} = 0.002$, $\sigma_{\max} = 80.0$, controls noise magnitude along the trajectory. During inference, we solve the ODE using a fifth-order Runge–Kutta integrator [33] with adaptive step size, ensuring accurate, stable transformation of initial noise into high-resolution H&E or multi-channel (e.g., DAPI) histological images.

We trained our models for 100 epochs using the AdamW optimizer [34] with a batch size of 96. The learning rate followed a cosine annealing schedule [35] with a minimum learning rate set to 1% of the initial value, helping the model converge more smoothly during later stages of training. All experiments were conducted on a single NVIDIA H100 GPU, with training times ranging around 12 hours per experiment (on full sample) requiring up to 78 GB of VRAM.

## 4 Results

### 4.1 Datasets

To curate our training data with high resolution H&E stained image and real single-cell level resolved spatial transcriptomics data, we used three large publicly available spatial transcriptomics datasets prepared with 10x Genomics' Xenium platform [36], all derived from Formalin-Fixed Paraffin-Embedded (FFPE) human melanoma samples. Among these samples, two were prepared using standard gene panels or with add-on custom gene targets, including around 300 genes. Another sample was from the Xenium Prime panel with 5000 targeting genes. For identification, we name these datasets $\text{Xenium}_{C1}$, $\text{Xenium}_{C2}$, and $\text{Xenium}_{P1}$. We collected 40X H&E stained images and aligned images with auxiliary staining such as DAPI and 18S (stain nucleus and cell boundaries respectively), which were used for cell segmentation by 10x Xenium Analyzer. Based on identified cell boundaries, we locate the cell at the center and extracted 256×256 pixels square-size image for consistency, with or without cell boundary mask. Only 126 genes are shared across the aforementioned samples, while the majority of genes remain substantially different due to various designs of Xenium panel. This results in gene expression data that are effectively collected from heterogeneous distributions, making the dataset a good fit to assess the generalizability of our models. We also extended experiments to 59 human Xenium samples from 12 organs in the HEST-1k dataset [37], totaling 1.6M paired patches.

Transcriptomics data were processed following standard protocols. Pre-identified Cells with unusually low or high gene counts were removed during quality check. Gene expression profiles were normalized and log-transformed to stabilize variance and reduce the influence of outliers. We further removed bottom 5% cells with the lowest total gene count to exclude low-quality or degraded cells. For single-cell modeling, we aligned and paired each individual cell's gene expression data with its corresponding image tile, including auxiliary channels when available. To

Table 1: Dataset Cellularity

|  | $\text{Xenium}_{C1}$ | $\text{Xenium}_{C2}$ | $\text{Xenium}_{P1}$ |
| --- | --- | --- | --- |
| Total patches | 9394 | 39334 | 13832 |
| Total cells | 106980 | 70178 | 137927 |
| B/Plasma cells | - | - | 3435 |
| Endothelial cells | 4123 | 4182 | 5110 |
| Epithelial cells | 11105 | 4203 | 2425 |
| Fibroblasts | 12091 | 9694 | 11003 |
| Macrophages | 2739 | 12088 | 15728 |
| Melanoma cells | 70539 | 33309 | 47423 |
| T cells | - | 15272 | 12871 |

simulate tissue-level heterogeneity and cell-cell interactions, we also created patch-level data using a sliding window approach with 256×256 pixel windows and 100-pixel overlap. Within each patch, we aggregated transcriptomic profiles from cells completely enclosed in the window to ensure accurate context matching. Cell types shown in Table 1 are identified by canonical cell type markers widely used by previous melanoma studies and further verified by differentially expressed gene and pathway analyses. To test model performance in the presence of class imbalance and potential catastrophic forgetting, we created subsets of the data, focusing solely on melanoma or non-melanoma cells,

which served as the basis for targeted ablation experiments. Signatures and differentially expressed genes used for cell type annotation can be found in Appendix A.

## 4.2 Quantitative Image Analysis

We benchmarked the GeneFlow architecture against a baseline diffusion-based generative model, which uses the same structure for the single- and multi-cell gene encoder, but differs in its image generation dynamics. Both models were trained separately on each of the spatial transcriptomics datasets. For each dataset, we performed 3-fold cross-validation to ensure robustness and generalizability of our results across tissue variations. To assess the quality of the generated histological images, we used three widely accepted evaluation metrics, each capturing different aspects of visual fidelity and biological plausibility. Structural Similarity Index Measure (SSIM) evaluates how perceptually similar the generated images are to the ground truth by comparing structural elements such as texture, contrast, and brightness. Higher SSIM scores indicate greater visual and structural resemblance. Fréchet Inception Distance (FID) [38] measures how close the overall distribution of generated images is to that of real histology images. It does this by comparing statistical summaries; specifically, the means and covariances of image features extracted by a pretrained neural network. Lower FID scores indicate greater realism and similarity to the real image distribution. Feature Distance in Inception Space quantifies the average difference between individual generated and real image pairs by comparing their features in the latent space of the same pretrained network. Unlike FID, which offers a global view, this metric focuses on localized, sample-by-sample feature differences.

We evaluated our rectified flow method against a diffusion baseline across three datasets, using both single-cell and multi-cell models trained and tested on all cell types (Table 2). Rectified flow consistently outperforms the baseline across all metrics, delivering substantially better image quality with FID scores 3-6 times lower. Single-cell models generally perform better than multi-cell ones, with the $\text{Xenium}_{C1}$ single-cell model achieving the best FID (20.73), suggesting stronger capture of intra-cellular gene expression morphology relationships. Notably, our model obtained comparable level of performance over all three evaluation metrics on multi-cell mode, which indicates our model's capability to learn tile level inter-cellular features and local tissue structures.

Table 2: Rectified Flow and Diffusion models trained and tested on all cell types

| Sample | Model | Rectified Flow | | | Diffusion | | |
| | | ↓ FID | ↑ SSIM± | ↓ FeatDist± | ↓ FID | ↑ SSIM± | ↓ FeatDist± |
|---|---|---|---|---|---|---|---|
| $\text{Xenium}_{P1}$ | multi | 34.31±5.65 | **0.23±0.11** | **13.41±2.19** | 213.60±17.20 | 0.18±0.19 | 17.90±2.56 |
| | single | **27.43±5.91** | 0.17±0.03 | 14.53±2.15 | 132.09±57.05 | 0.20±0.07 | 17.11±2.22 |
| $\text{Xenium}_{C1}$ | multi | 47.95±7.38 | **0.28±0.10** | **14.50±2.80** | 189.08±13.40 | 0.30±0.18 | 19.31±2.47 |
| | single | **20.73±8.45** | 0.24±0.04 | 14.90±2.55 | 171.06±81.98 | 0.22±0.07 | 18.53±2.81 |
| $\text{Xenium}_{C2}$ | multi | 45.50±4.10 | **0.40±0.09** | **15.61±2.26** | 208.86±40.93 | 0.24±0.15 | 20.46±2.75 |
| | single | **42.61±4.50** | 0.35±0.06 | 15.65±2.28 | 119.22±40.21 | 0.36±0.12 | 18.24±2.47 |

Table 3: Rectified Flow model trained and tested on melanoma and non-melanoma cells

| Sample | Model | Melanoma Cells | | | Non-Melanoma Cells | | |
| | | ↓FID± | ↑ SSIM± | ↓ FeatDist± | ↓ FID | ↑ SSIM± | ↓ FeatDist± |
|---|---|---|---|---|---|---|---|
| $\text{Xenium}_{P1}$ | multi | 181.72±3.12 | **0.36±0.21** | 15.68±2.99 | 264.06±19.63 | 0.24±0.08 | 18.42±3.18 |
| | single | **14.18±1.86** | 0.17±0.02 | **13.95±2.01** | 47.01±14.81 | 0.18±0.04 | 14.82±2.23 |
| $\text{Xenium}_{C1}$ | multi | 104.50±2.99 | **0.37±0.18** | **13.71±2.48** | 348.21±1.82 | 0.35±0.10 | 20.43±3.49 |
| | single | **22.86±1.98** | 0.22±0.03 | 14.12±2.48 | 96.28±51.64 | 0.21±0.03 | 16.25±2.43 |
| $\text{Xenium}_{C2}$ | multi | **24.43±0.17** | **0.52±0.05** | **13.94±2.08** | 272.79±18.30 | 0.39±0.06 | 20.12±2.21 |
| | single | 46.30±14.44 | 0.37±0.05 | 15.74±2.31 | 65.24±6.77 | 0.40±0.05 | 15.97±2.20 |

We further compared generation performance between models trained on datasets containing all cell types versus those trained exclusively on either melanoma or non-melanoma cells (Table 3). Models trained on non-melanoma cells generally showed degraded performance, likely due to the heterogeneity of immune cell types, imbalance in cell abundance, and their non-uniform spatial distribution. Additionally, because non-melanoma cells often co-occur with melanoma cells in tumor regions, training data labeled as non-melanoma may still contain partial melanoma features, which can confuse the model's gene-to-image mapping. In contrast, models trained on more curated and homogeneous melanoma-only datasets performed comparably or better than models trained on all cell

types. This highlights the model's ability to learn precise, cell type-specific morphological features from gene expression data when provided with sufficiently clean and targeted training samples.

To assess generalization, we train on one dataset and test on another (Table 4). Despite limited gene panel overlap (126 shared genes), rectified flow maintains strong cross-dataset performance. Models trained on $Xenium_{C1}$ and tested on P1 yield the best results (FID: 67.00 single-cell, 79.86 multi-cell), with stable SSIM and feature distance scores, demonstrating robust transferability and ability to learn dataset-agnostic gene morphology mappings. Benchmarking details and auxiliary experiments can be found in Appendix B.1.

| Train Sample | Test Sample | Model | ↓ FID± | ↑ SSIM± | ↓ FeatDist± |
|---|---|---|---|---|---|
| $Xenium_{P1}$ | $Xenium_{C1}$ | multi | **89.16±10.22** | 0.30±0.17 | **16.11±2.35** |
| | $Xenium_{C1}$ | single | 90.88±2.01 | 0.24±0.05 | 16.13±2.04 |
| | $Xenium_{C2}$ | multi | 127.79±13.28 | **0.38±0.11** | 17.27±2.33 |
| | $Xenium_{C2}$ | single | 144.49±3.32 | 0.34±0.09 | 17.40±2.07 |
| $Xenium_{C1}$ | $Xenium_{P1}$ | multi | 79.86±11.45 | 0.30±0.18 | **15.80±2.33** |
| | $Xenium_{P1}$ | single | **67.00±11.61** | 0.20±0.07 | 15.91±2.06 |
| | $Xenium_{C2}$ | multi | 112.25±11.02 | **0.37±0.12** | 17.18±2.15 |
| | $Xenium_{C2}$ | single | 98.78±12.09 | 0.36±0.09 | 17.20±2.09 |
| $Xenium_{C2}$ | $Xenium_{P1}$ | multi | 148.15±0.72 | 0.29±0.19 | **17.59±2.01** |
| | $Xenium_{P1}$ | single | 164.35±7.36 | 0.19±0.07 | 18.02±1.93 |
| | $Xenium_{C1}$ | multi | 146.78±29.10 | **0.34±0.17** | 18.36±2.36 |
| | $Xenium_{C1}$ | single | **145.44±31.13** | 0.24±0.06 | 18.21±2.31 |

Table 4: Rectified Flow model cross-dataset performance evaluation

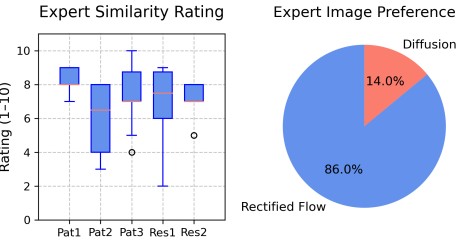

Figure 3: Evaluation by 3 ACVP board certified pathologists and 2 residents.

Single-cell models (Tables 2-3) outperform multi-cell models due to both methodological and biological factors. Unlike sliding-window multi-cell generation, single-cell patches are centered on valid cells with transcript overlap above a threshold and use spatial weighting that decreases toward patch margins. This emphasizes target-cell morphology while reducing peripheral effects. Patch sizes accommodate full cells and variability, yielding more homogeneous patches. Single-cell conditioning provides focused gene context, whereas multi-cell patches mix heterogeneous cell types, complicating gene-to-morphology mapping. In melanoma, the diverse tumor-immune-stromal microenvironment further degrades multi-cell performance in boundary resolution and texture consistency.

We evaluate performance using histopathology specific metrics derived from the UNI2-h foundational model [39] in Table 5. These include UNI2-h FID for pathology-specific image quality, UNI2-h embedding similarity for comparing feature distributions, and nuclear morphometric similarity quantifying circularity, eccentricity, and solidity of segmented nuclei. We further assess spatial energy similarity from gray-level co-occurrence matrices, as well as spatial complexity and feature magnitude from UNI2-h embeddings to capture tissue structure.

Table 5: Domain-specific evaluation metrics for C1 model

| Metric | C1 Multi Diff. | C1 Multi Rect. | C1 Single Diff. | C1 Single Rect. |
|---|---|---|---|---|
| *Image Quality Metrics* ↓ | | | | |
| FID Overall UNI2-h | 404.06 | **87.96** | 405.98 | **39.27** |
| Inception Feat. Dist. | 23.05±1.86 | **16.72±1.64** | 20.18±1.61 | **14.50±2.52** |
| *Biological Feature Similarity* ↑ | | | | |
| UNI2-h Embedding Sim. | 0.967±0.007 | **0.979±0.004** | 0.969±0.007 | **0.983±0.003** |
| Nuclear Circularity Sim. | 0.835±0.043 | **0.844±0.037** | 0.839±0.049 | **0.874±0.028** |
| Nuclear Eccentricity Sim. | 0.869±0.046 | **0.954±0.015** | 0.880±0.039 | **0.964±0.012** |
| Nuclear Solidity Sim. | 0.714±0.048 | **0.888±0.036** | 0.721±0.077 | **0.867±0.028** |
| *Spatial Feature Metrics* ↑ | | | | |
| Spatial Energy Sim. | – | **0.283±0.083** | – | **0.678±0.176** |
| Spatial Complexity Sim. | 0.167±0.138 | **0.506±0.080** | 0.130±0.111 | **0.571±0.085** |
| Spatial Feat. Magnitude Sim. | 0.167±0.143 | **0.514±0.077** | 0.135±0.119 | **0.571±0.084** |

These metrics provide finer insights than standard vision metrics, diagnosing limitations in boundary definition and texture fidelity. In table 6 we benchmark against the diffusion model along with an implementation of conditional UNet baseline with our gene encoder trained using MSE and perceptual losses. Ablations show the transformer-based RNA encoder consistently outperforms simpler encoders across all metrics. While dropping components causes modest performance declines, each contributes uniquely to interpretability, supporting identification of important genes and their relationships. The encoder's complexity is justified, balancing strong performance with interpretable gene-morphology links.

To explicitly preserve the spatial organization of cells in generated histopathology images, we introduced a spatial graph loss that enforces consistency in local tissue architecture. We propose two complementary approaches, (1) a segmentation-based method that models nuclear morphology and spatial relationships and (2) a fast alternative gradient-based method that captures texture patterns through local image derivatives and neighborhood similarity. Both approaches construct kNN graphs in spatial coordinates and penalize discrepancies in local appearances between generated and ground

Table 6: C1 Single-Cell model baseline comparison and ablation study. Including larger HEST-1k dataset for Multi-Cell model

| Metric | UNet (MSE) | Diffusion | Rectified | -Gene Att. | -Gene Att./Rel. | -Gene/-Multi Att. | HEST-1k |
|---|---|---|---|---|---|---|---|
| *Image Quality Metrics* ↓ | | | | | | | |
| FID Overall UNI2-h | 525.51 | 405.98 | 39.27 | **26.46** | 27.25 | 29.50 | 62.54 |
| Inception Feat. Dist. | 21.76±1.29 | 20.18±1.61 | **14.50±2.52** | 14.89±2.52 | 15.02±2.70 | 15.39±2.68 | 15.26±2.70 |
| *Biological Feature Similarity* ↑ | | | | | | | |
| UNI2-h Embedding Sim. | 0.964±0.004 | 0.969±0.007 | 0.983±0.003 | **0.991±0.003** | 0.990±0.003 | 0.990±0.003 | 0.974±0.004 |
| Nuclear Circularity Sim. | 0.659±0.038 | 0.839±0.049 | 0.874±0.028 | **0.949±0.022** | 0.941±0.021 | 0.947±0.020 | 0.924±0.027 |
| Nuclear Eccentricity Sim. | 0.655±0.024 | 0.880±0.039 | 0.964±0.012 | **0.959±0.013** | 0.954±0.015 | 0.957±0.017 | 0.955±0.014 |
| Nuclear Solidity Sim. | 0.479±0.037 | 0.721±0.077 | 0.867±0.028 | **0.950±0.020** | 0.943±0.019 | 0.949±0.020 | 0.921±0.028 |
| *Spatial Feature Metrics* ↑ | | | | | | | |
| Spatial Energy Sim. | 0.018±0.034 | – | 0.678±0.176 | **0.723±0.130** | 0.735±0.055 | 0.719±0.048 | 0.744±0.049 |
| Spatial Complexity Sim. | 0.099±0.060 | 0.130±0.111 | 0.571±0.085 | **0.749±0.069** | 0.704±0.072 | 0.721±0.067 | 0.641±0.061 |
| Spatial Feat. Magnitude Sim. | 0.112±0.061 | 0.135±0.119 | 0.571±0.084 | **0.759±0.068** | 0.718±0.068 | 0.735±0.069 | 0.635±0.059 |

truth images. The spatial loss is gradually introduced during training which regularize that generated images maintain biologically plausible cell arrangements and tissue microarchitecture, improving both visual fidelity and downstream biological interpretability. Details can be found in appendix E.

## 4.3 Gene Importance Analysis

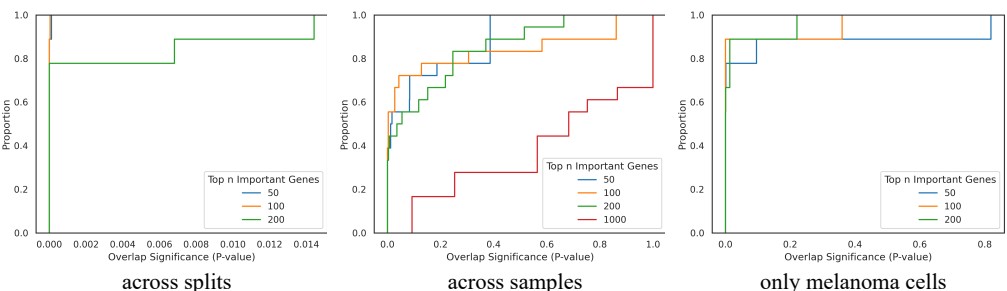

Figure 4: Overlap significance of influential genes

To interpret gene expression-phenotype relationships learned by our generative model, we performed a gradient-based sensitivity analysis to quantify each gene's influence on cellular morphology. Specifically, we computed the partial derivatives of the predicted velocity's squared L2-norm with respect to individual gene expression levels across multiple stochastic evaluations and generative timesteps. This allowed us to rank genes based on their impact on morphological outcomes, providing potential insight into how high-dimensional gene expression patterns shape complex visual phenotypes. These importance scores can guide hypotheses about gene function and regulatory mechanisms, and help prioritize candidates for downstream experimental validation.

We evaluated the consistency of the gene sets with the highest importance scores by counting the overlap of the top 50, 100, and 200 important genes across multiple model variants, data splits, and datasets with differing cellular composition and gene panels. The statistical significance of the overlapping genes was assessed using a hypergeometric test. As shown in Figure 4, we consistently observed statistically significant overlaps in influential genes across different splits, with 60 to 80% of comparisons showing substantial agreement, even though only 126 genes were shared across the three gene panels. This highlights our model's ability to generalize across sample-specific gene panels. When comparing models trained on all cell types versus melanoma-only cells (right panel), over 80% of comparisons showed significant gene overlap. This is consistent with melanoma cells being the dominant population in all datasets. In contrast, there was no significant overlap between influential genes from melanoma-only and non-melanoma-only models, reflecting distinct underlying gene-phenotype relationships in these cell populations.

To further validate biological relevance, we performed a gene set enrichment analysis [40] on the 34 shared genes out of top 50 influential genes between two samples with standard panels and custom add-on ($Xenium_{C1}$ and $Xenium_{C2}$). The results (see Appendix C) revealed that EMT and extracellular

matrix organization pathways are significantly enriched. These pathways play critical roles in melanoma progression, with EMT-like phenotype switching contributing to metastatic potential and ECM remodeling facilitating invasion. Our model independently prioritized genes within these pathways without any explicit biological knowledge encoded in its architecture; demonstrating the model's ability to successfully capture the fundamental relationship between gene expression and cellular morphology visible in histopathological images.

## 4.4 Qualitative Image Analysis

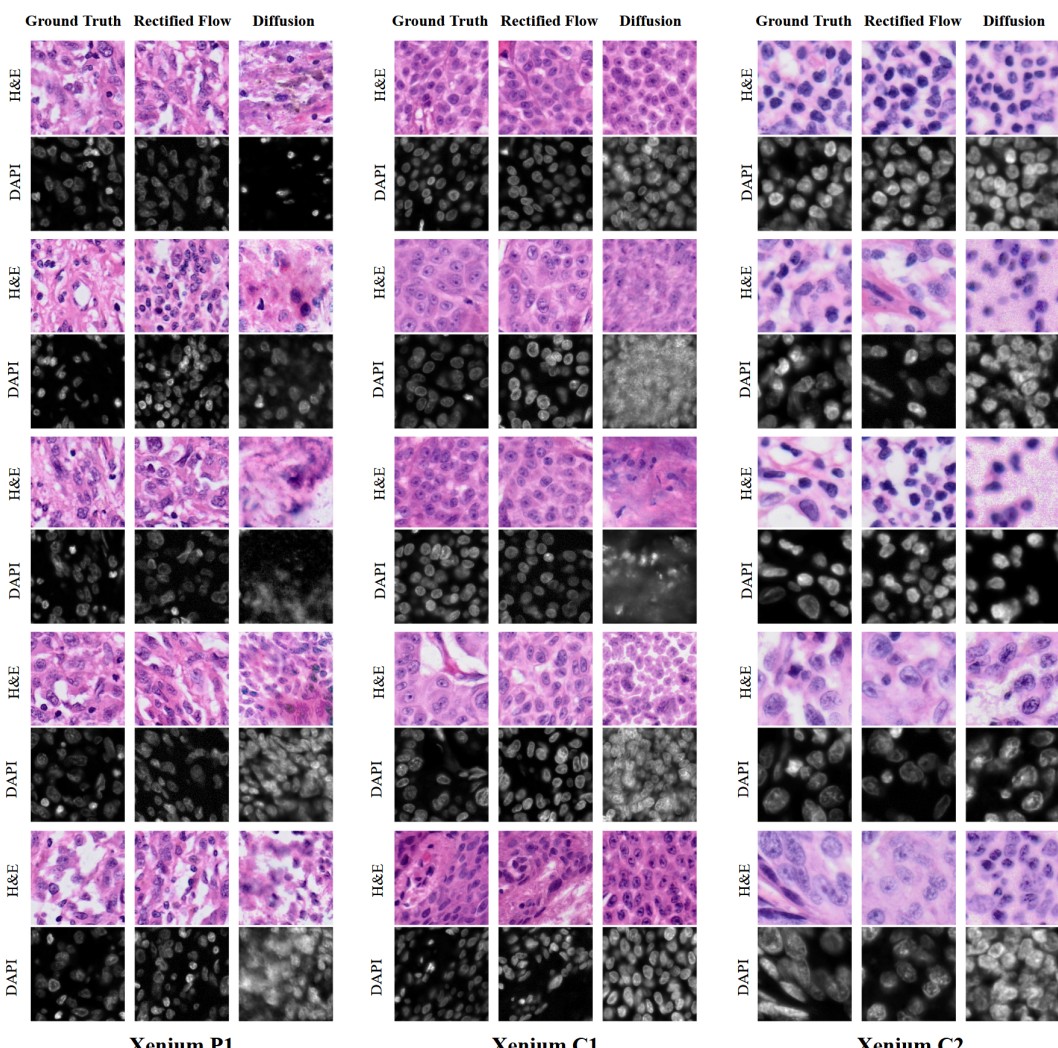

Figure 5: Comparison of ground-truth and generated images for Rectified Flow and Diffusion models

Figure 5 shows the ground-truth and generated images through the rectified flow and diffusion models from three different datasets. Rectified flow model clearly outperforms the diffusion model by producing more accurate H&E and DAPI images of cellular and tissue morphology given gene expression data. The generated cellular, nuclear and nucleoli morphologies are visually consistent with ground truth. Further generated image results from all experiments can be found in Appendix B.3. We conducted a human evaluation study with three ACVP board certified pathologists and two residents, using similarity and preference tasks. In the similarity task, pathologists were shown 20 pairs of rectified flow-generated and ground truth images from the test set, and asked to blindly rate their similarity on a scale of 1-10. In the preference task, they were shown 20 pairs of generated images one from the rectified flow model and one from the diffusion model and asked to blindly choose which they preferred for cell and tissue classification clarity. As shown in Figure 3, all

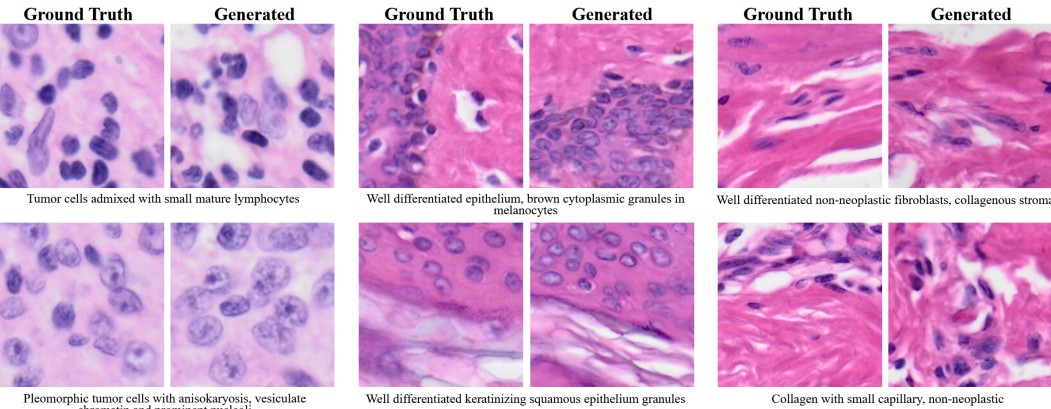

| Ground Truth | Generated | Ground Truth | Generated | Ground Truth | Generated |

Tumor cells admixed with small mature lymphocytes | Well differentiated epithelium, brown cytoplasmic granules in melanocytes | Well differentiated non-neoplastic fibroblasts, collagenous stroma

Pleomorphic tumor cells with anisokaryosis, vesiculate chromatin and prominent nucleoli | Well differentiated keratinizing squamous epithelium granules | Collagen with small capillary, non-neoplastic

Figure 6: Ground-truth and generated H&E image with diagnosis by ACVP certified pathologists.

pathologists rated similarity above a median of 6, and preferred the image generated using the rectified flow model over diffusion in 86% of the cases.

To demonstrate a diagnostic application of GeneFlow, Figure 6 presents representative examples of generated histology patches alongside their corresponding ground-truth images from both tumor (melanoma) and healthy skin tissues. ACVP board certified pathologist were asked to provide diagnoses based solely on the generated cellular and microenvironmental morphology. Neoplastic regions characterized by clonal cellular proliferation include both benign and malignant tumor cells, while non-neoplastic regions encompass normal epithelium, inflammatory infiltrates, fibroblastic stroma, and reactive tissue changes. The generated images faithfully reproduce key diagnostic features such as pleomorphic nuclei, keratinizing squamous epithelium, and collagenous stroma, enabling pathologist to reach consistent interpretations with high confidence relative to ground truth. These results highlight GeneFlow's potential to synthesize diagnostically coherent tissue morphologies from transcriptomic data, supporting minimally invasive, transcriptome-guided pathology.

## 5  Discussion

GeneFlow bridges the gap between transcriptomics and cellular morphology by introducing a novel framework that translates gene expression profiles into realistic histopathological images. Leveraging rectified flow dynamics, our method consistently outperforms diffusion-based alternatives, as demonstrated by both quantitative benchmarks and pathologist assessments. By effectively modeling the complex relationship between gene expression and histological features, GeneFlow enables new opportunities for minimally invasive diagnostics and virtual tissue reconstruction. The observed performance gap between single-cell and multi-cell models also offers valuable insights into model scalability. While the single-cell variant currently achieves higher accuracy, this highlights a promising direction for improving the multi-cell architecture to better capture intercellular dynamics and tissue-level structure.

While GeneFlow offers a strong foundation, several limitations highlight important directions for future work. Our current implementation operates on 256×256 pixel tiles, serving as a proof-of-concept that can be extended to whole-slide image synthesis using hierarchical generation or sliding-window strategies with boundary-aware stitching. Additionally, our reliance on Xenium data featuring true single-cell resolution limits compatibility with a broader range of spatial transcriptomics datasets that offer only near single-cell resolution. Scaling to datasets with more comprehensive gene panels may also pose challenges, as the model must handle higher-dimensional input. This could be addressed by integrating foundational models trained on single-cell or tissue-level gene expression.

Finally, our current cell alignment method treats cell inclusion as binary. Incorporating weighted contributions based on partial overlap would enable more nuanced modeling of cell context. These limitations point to natural and promising extensions of our framework. Future work will also explore next-generation generative models and large-scale training on whole-slide images to further advance the mapping between gene expression and visual cellular phenotypes at high resolution.

## Acknowledgement

This work was supported in part by the Walther Cancer Foundation and the Purdue Institute for Cancer Research (Grant P30CA023168).

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

# A   Datasets Details

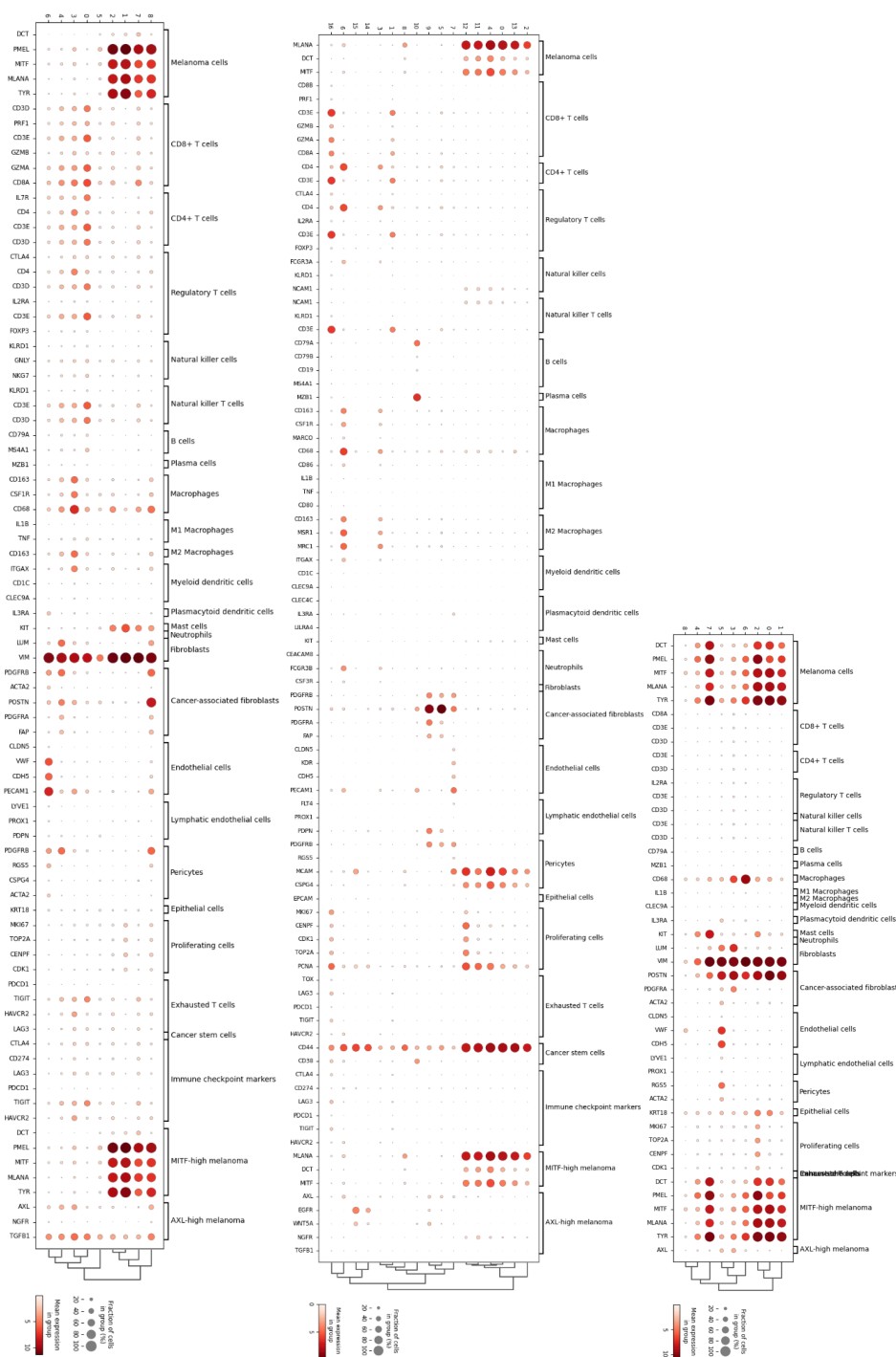

Figure 7: Marker genes expressions of major cell types in melanoma samples. Left: Xenium$_{C2}$.; middle: Xenium$_{P1}$; right: Xenium$_{C1}$.

Xenium data includes two major modalities that provide complementary biological information. The image modality consists of fluorescence microscopy images that capture individual RNA molecules at subcellular resolution, along with H&E (hematoxylin and eosin) stained images that reveal tissue morphology and cellular architecture. The transcriptomics modality provides spatially resolved gene

expression data for hundreds of targeted genes, enabling us to map transcript locations and quantify expression levels within specific tissue regions and individual cells. We extracted cell centroid from pre-processed melanoma Xenium data, locating cell centroid at the center of image with size of 256 pixels. We do not mask cell by its boundary due to 1) cells may overlap with each other which may cause inaccurate fluorescence-based cell boundary identification 2) actual cell size can be affected by the sample preparation that cells more densely collapsed together or loosely isolated may lead to abnormal cell size and 3) without masking our model can learn more than intrinsic cell features but the environments where that cell resides via the environment information (neighboring cells, etc.) provided in the unmasked image, which is key to single-cell level accurate image reconstruction for precise pathological diagnosis.

For each sample, we further processed the transcriptomics data and clustering the cells based on their transcriptomes profiles. We annotated cell types referring to marker genes from [41, 42], Marker genes expressions are visualized in the dot plots.

# B  Experiments

## B.1  Quantitative Results

| Sample | Model | FID | SSIM$\pm$ | Feature Dist$\pm$ |
|---|---|---|---|---|
| Xenium$_{P1}$ | multi | 105.54$\pm$6.65 | 0.28$\pm$0.13 | 13.88$\pm$2.38 |
| Xenium$_{P1}$ | single | 32.43$\pm$16.75 | 0.17$\pm$0.03 | 14.22$\pm$2.08 |
| Xenium$_{C1}$ | multi | 63.41$\pm$3.83 | 0.33$\pm$0.13 | 12.61$\pm$2.69 |
| Xenium$_{C1}$ | single | 25.47$\pm$16.10 | 0.23$\pm$0.03 | 14.56$\pm$2.54 |
| Xenium$_{C2}$ | multi | 41.45$\pm$6.80 | 0.51$\pm$0.05 | 14.84$\pm$2.13 |
| Xenium$_{C2}$ | single | 21.85$\pm$5.76 | 0.49$\pm$0.05 | 15.10$\pm$2.22 |

Table 7: Rectified Flow model trained on all cell types but tested only on melanoma cells

| Sample | Model | FID | SSIM$\pm$ | Feature Dist$\pm$ |
|---|---|---|---|---|
| Xenium$_{P1}$ | multi | 50.09$\pm$11.37 | 0.32$\pm$0.16 | 13.23$\pm$2.41 |
| Xenium$_{P1}$ | single | 37.08$\pm$4.61 | 0.17$\pm$0.03 | 14.83$\pm$2.19 |
| Xenium$_{C1}$ | multi | 94.79$\pm$4.82 | 0.41$\pm$0.14 | 16.03$\pm$3.04 |
| Xenium$_{C1}$ | single | 34.59$\pm$15.01 | 0.25$\pm$0.05 | 15.74$\pm$2.38 |
| Xenium$_{C2}$ | multi | 54.18$\pm$2.97 | 0.37$\pm$0.09 | 15.85$\pm$2.34 |
| Xenium$_{C2}$ | single | 51.38$\pm$12.59 | 0.27$\pm$0.04 | 15.75$\pm$2.30 |

Table 8: Rectified Flow model trained on all cell types but tested only on non-melanoma cells

We conducted a series of evaluation experiments using Rectified Flow models trained on all cell types but tested under three specific conditions: (1) only on melanoma cells, (2) only on non-melanoma cells, and (3) separately on each non-melanoma cell type. Tables 7–9 report the corresponding evaluation metrics across different samples and configurations. Models tested exclusively on melanoma cells (Table 7) exhibited strong performance, particularly in the single-cell conditioned setting. Notably, while multi-cell conditioning resulted in marginally lower FID scores in some cases, it consistently led to higher SSIM values, indicating better structural similarity. This suggests that multi-cell generation better preserves spatial structure when the sample contains a mixture of cell types, as is typical even in melanoma-dominated tissues. Evaluation on non-melanoma cells (Table 8) showed that models trained on all cell types maintained good generalization, though the variability in FID and SSIM scores increased across samples. Again, the multi-cell conditioned models tended to produce more structurally consistent images (higher SSIM), whereas single-cell conditioned models sometimes yielded lower FID scores, indicating perceptually closer images. The most granular analysis testing on individual non-melanoma cell types (Table 9) revealed a slight drop in performance across all metrics. This decline can likely be attributed to the reduced diversity in conditioning data: conditioning on a single cell type eliminates contextual transcriptomic variation that may assist the model in disambiguating cell-specific morphological features. In particular, cell types such as epithelial and endothelial cells demonstrated higher FID and lower SSIM, suggesting that their complex spatial

| Gene/Cell Type (P1) | FID | SSIM± | Feature Dist± |
| --- | --- | --- | --- |
| B/Plasma | 59.02±8.59 | 0.17±0.03 | 14.69±2.01 |
| Endo | 49.75±2.52 | 0.17±0.03 | 14.72±2.05 |
| Epith | 191.30±9.21 | 0.21±0.04 | 17.60±1.97 |
| Fibro | 53.22±9.68 | 0.18±0.03 | 15.02±2.15 |
| Mono/Mac | 32.28±3.64 | 0.17±0.03 | 14.47±2.04 |
| T cells | 37.63±3.13 | 0.16±0.03 | 14.57±2.04 |

| Gene/Cell Type (C1) | FID | SSIM± | Feature Dist± |
| --- | --- | --- | --- |
| Endo | 69.82±17.87 | 0.26±0.05 | 16.40±2.29 |
| Epith/Kera | 51.14±28.75 | 0.27±0.05 | 15.68±2.48 |
| Fibro/Stroma | 39.57±13.39 | 0.24±0.04 | 15.76±2.25 |
| Macro/Myeloid | 40.76±3.41 | 0.23±0.04 | 14.83±2.27 |

| Gene/Cell Type (C2) | FID | SSIM± | Feature Dist± |
| --- | --- | --- | --- |
| Endo | 80.71±10.73 | 0.29±0.04 | 16.36±2.37 |
| Epith/Kera | 71.04±10.03 | 0.42±0.06 | 16.47±2.42 |
| Fibro | 61.50±11.25 | 0.29±0.05 | 15.98±2.30 |
| Macro/Mono | 59.19±12.77 | 0.29±0.04 | 15.85±2.18 |
| T cells | 56.93±16.43 | 0.24±0.03 | 15.11±2.19 |

Table 9: Rectified Flow model tested on non-melanoma cell types in (top) $P_1$, (middle) $C_1$, and (bottom) $C_2$.

morphologies or transcriptional profiles are harder to reconstruct accurately under single-cell-type conditioning. Overall, our results indicate that while the model performs robustly across all settings, multi-cell-type conditioning offers a tangible advantage, especially for maintaining spatial fidelity in heterogeneous tissues. The detailed evaluation metrics for each configuration are reported in Tables 7, 8, and 9.

## B.2 Ablation Study

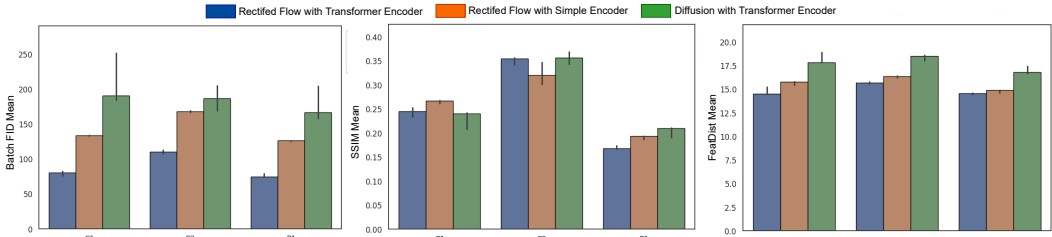

Figure 8: Ablation study metric comparison between Rectfied Flow with a transformer encoder vs Rectfied Flow with a simple encoder vs Diffusion with a transformer encoder.

To assess the effectiveness of our transformer-based RNA encoder, we conducted an ablation study comparing three configurations: (1) Rectified Flow with a transformer RNA encoder, (2) Rectified Flow with a simple RNA encoder, and (3) a standard diffusion model with a transformer encoder. The simple RNA encoder replaces attention mechanisms, residual blocks, and feature gating with a basic stack of fully connected linear layers to encode gene expression data. Figure 8 presents a comparative evaluation across three key metrics FID, SSIM, and Feature Distance on three representative samples (C1, C2, and P1). Models using the transformer-based RNA encoder consistently outperformed those with the simple encoder, particularly in terms of FID and Feature Distance. This demonstrates that incorporating attention and residual connections facilitates better conditioning on gene expression, leading to more realistic and feature-faithful image generation. Interestingly, when comparing Rectified Flow and Diffusion models both using transformer encoders, Rectified Flow generally achieved lower FID and Feature Distance scores, especially on C1 and P1, indicating superior visual quality and fidelity. However, Diffusion models showed competitive or slightly better SSIM on C2 and P1, suggesting that while they may achieve better pixel-wise similarity, Rectified Flow captures global image realism more effectively. These results validate the importance of architectural choices in the encoder design, highlighting the robustness and effectiveness of the transformer-based encoder in the context of spatial transcriptomic image synthesis.

## B.3 Qualitative Results

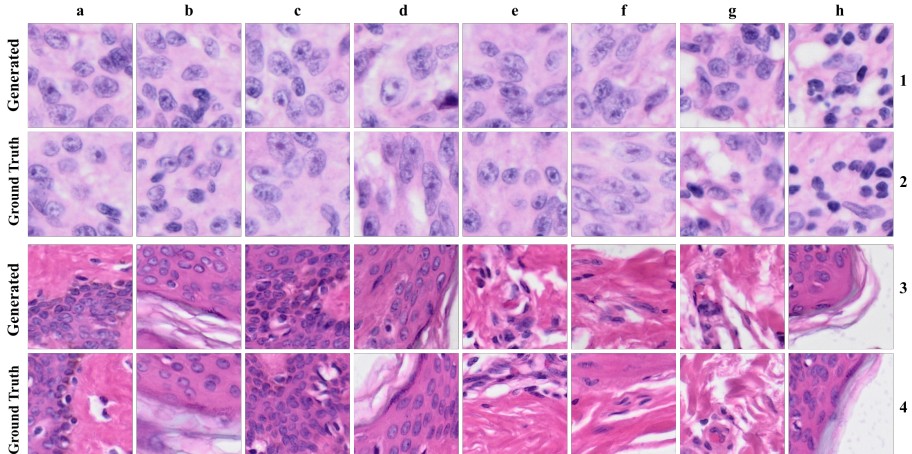

Figure 9: Extra examples of neoplastic tumor cells (Top) and non-neoplastic cells (Bottom), diagnosed by ACVP board certified pathologist.

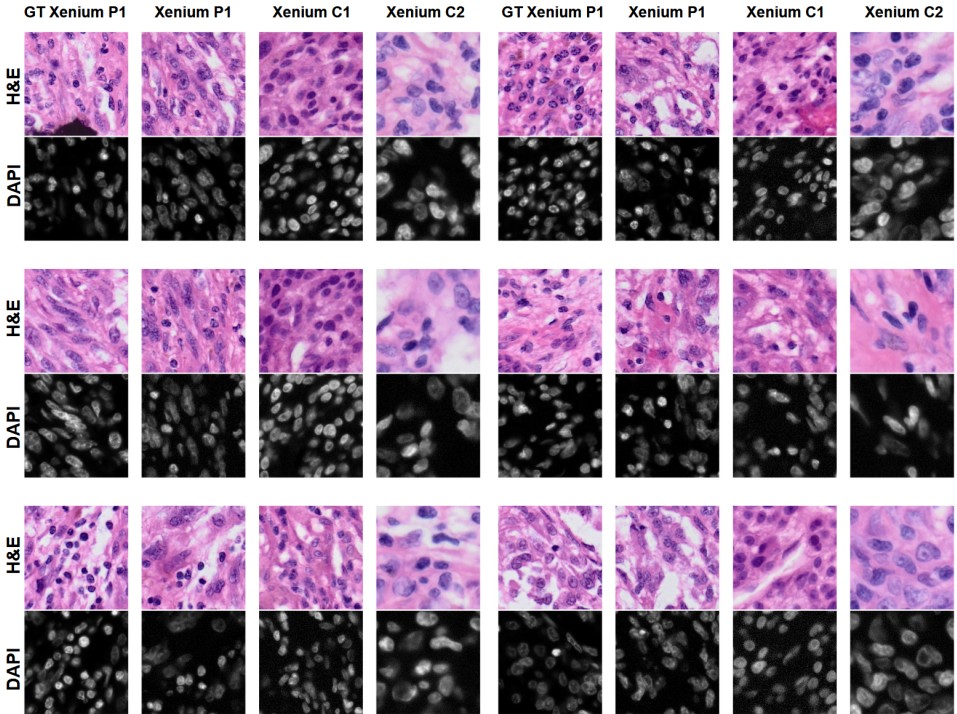

Figure 10: Comparison of ground-truth and generated images for Rectified Flow model trained on Xenium P1 and tested on Xenium P1, Xenium C1, Xenium C2. There are 2 columns with 3 rows.

Figures 10, 11, 12 show comparison of ground truth and generated images for the cross dataset evaluation task. This task corresponds to the results in Table 4. Here the Rectified Flow model is trained on the 3 datasets separately and tested on the held out datasets. These models were trained on the 126 overlapping shared genes.

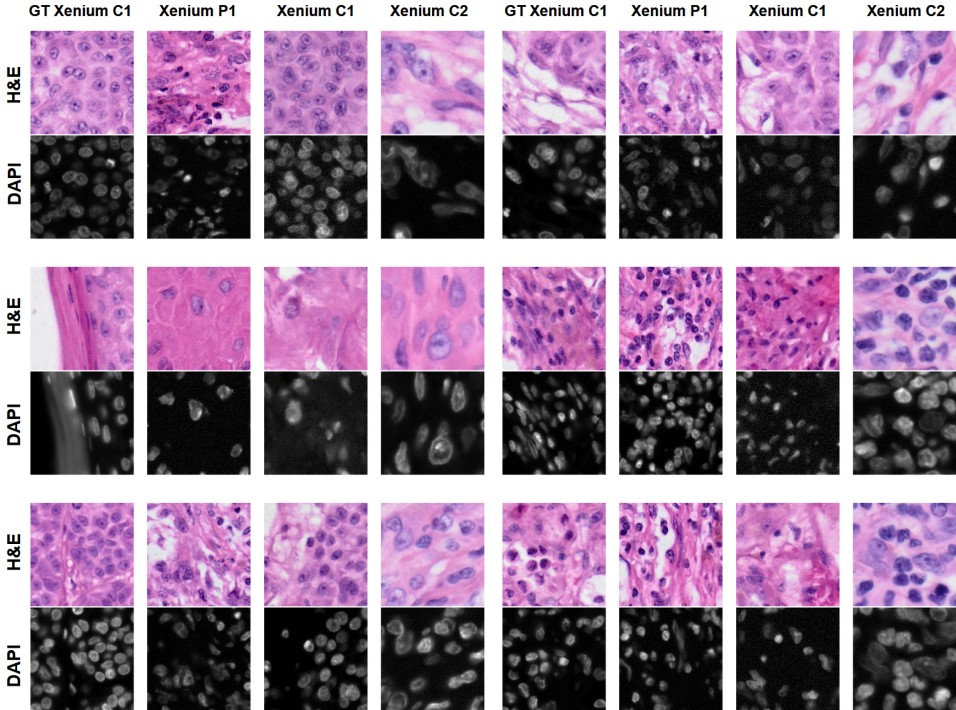

Figure 11: Comparison of ground-truth and generated images for Rectified Flow model trained on Xenium C1 and tested on Xenium P1, Xenium C1, Xenium C2. There are 2 columns with 3 rows.

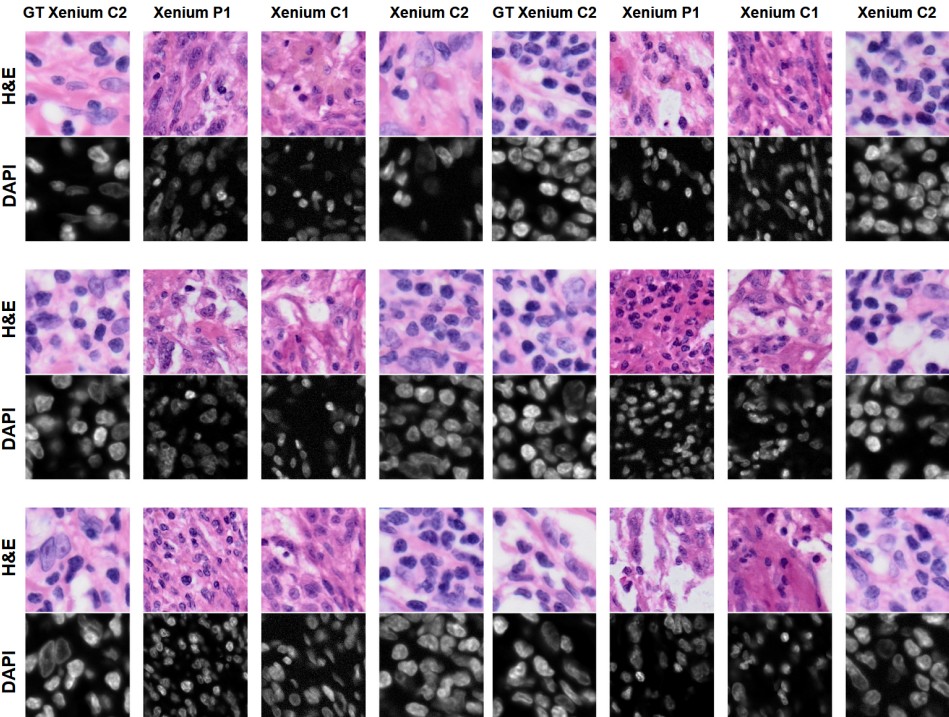

Figure 12: Comparison of ground-truth and generated images for Rectified Flow model trained on Xenium C2 and tested on Xenium P1, Xenium C1, Xenium C2. There are 2 columns with 3 rows.

# C Gene Influence Analysis

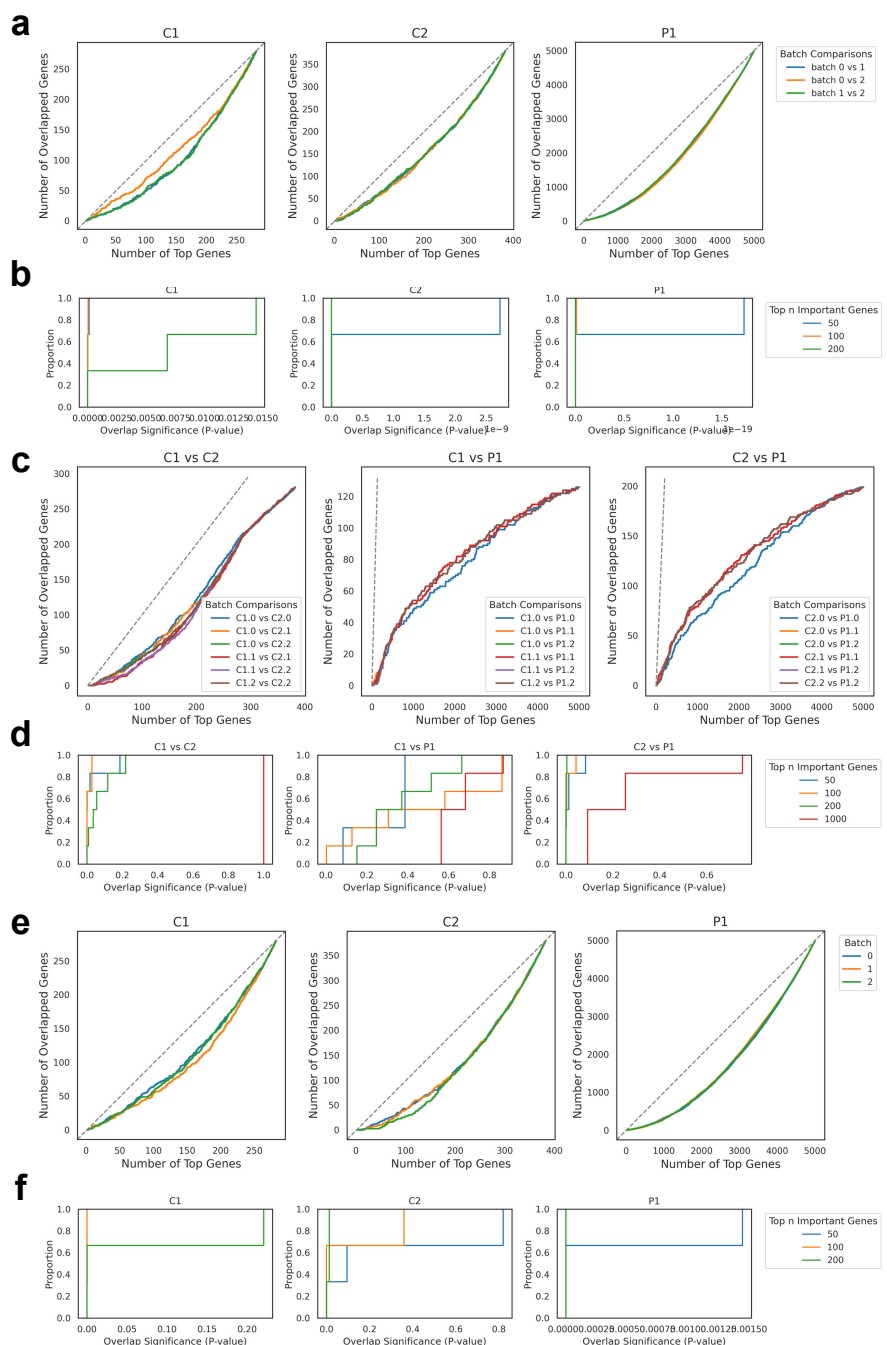

Figure 13: **Line charts and cumulative density plots for influential gene analysis. a, c, e.** Line charts showing the number of overlapping genes as a function of the total number of top influential genes examined. The dashed diagonal line indicates perfect overlap, where the number of overlaps equals the number of top genes considered. **b, d, f.** Cumulative density plots of p-values for the overlap of top influential genes under varying thresholds (top 50, 100, 200, and 1000 genes when Xenium $P_1$ is included). The x-axis shows p-values; the y-axis shows cumulative density. Different colors indicate different thresholds. **a, b.** Comparison across cross-validation folds. **c, d.** Comparison across biological samples. **e, f.** Comparison between models trained on all cell types vs. models trained only on melanoma cells.

We evaluated gene importance with respect to gene attention map as well as gradient flow analysis from all experiments, with respect to consistency across folds, consistency across models trained on different samples, and consistency across models trained on all cell types versus models trained on only melanoma cells. Our experiments demonstrated overall very high consistency with significant overlap across top important/influential genes from across folds evaluation scenario13(a,b), with p-value much less than 0.05 from one-tailed (greater) fisher-exact, regardless evaluating overlap with top 50, 100 or 200 influential genes. It indicated our model can stably capture marker genes contributing to cellular morphology during generation.

When comparing models trained on one sample while evaluated on unseen samples13(c,d), we observed high overlap between models trained on one of the two samples with standard gene panels and custom add-on and tested on the other, which is as expected since they share a major portion of target genes. All test cases with top 100 influential genes are statistically significant (p-value less than 0.05). Over 80% cases with top 50 have significant overlap.

Since the majority of cells from our curated dataset are melanoma cells, we expect our model to focus more on genes that are more important to picture melanoma cell morphology. Our most influential gene analysis revealed that there's significant overlap of top important genes between model trained on all cell types and model trained on only melanoma cells, demonstrating that our model's successfully focused on melanoma cells out of all cell types. This also matched the better metrics over evaluation against melanoma cells compared to against each other cell type from quantitative results analyses.

### C.1 Enrichment gene-sets/pathways from top influential genes

We evaluated top influential genes from comparisons across datasets, where we found highest consistency lying in sample $C_1$ and $C_2$. We selected shared genes from top 50 most influential genes from both datasets, sorted by average importance score, and ran gene-set enrichment analysis with Enrichr[40]. We found epithelial mesenchymal transition pathway is mostly enriched, along with KRAS signaling pathway, angiogenesis, apoptosis, myogenesis and coagulation which were identified to be closely relevant to melanoma development and progression by previous studies. This further demonstrates our model's capability on identifying disease relevant genes during generation cell/tissue images guided by transcriptomics. Details of significantly enriched pathways are attached as follows.

| Term | P-value | Adj. P-value | Odds Ratio | Combined Score | Genes |
|---|---|---|---|---|---|
| Epithelial Mesenchymal Transition | 2.09E-05 | 5.02E-04 | 17.48 | 188.37 | POSTN;LUM;MMP2;IGFBP2;MGP |
| KRAS Signaling Dn | 3.56E-04 | 4.27E-03 | 13.45 | 106.80 | TFAP2B;KRT15;IGFBP2;KRT5 |
| Angiogenesis | 1.70E-03 | 1.36E-02 | 36.64 | 233.56 | POSTN;LUM |
| Apoptosis | 2.55E-03 | 1.53E-02 | 12.13 | 72.44 | CCND1;LUM;MMP2 |
| Myogenesis | 4.69E-03 | 2.25E-02 | 9.71 | 52.07 | ACHE;APOD;CRYAB |
| Coagulation | 2.30E-02 | 9.18E-02 | 9.11 | 34.39 | C1QA;MMP2 |

Table 10: Gene set enrichment analysis of the most influential genes identified by our model. The table shows the top 10 enriched pathways ranked by adjusted p-value.

# D Model Architecture

## D.1 RNA Encoder

The RNA Encoder is a deep neural encoder designed to transform gene expression profiles from multiple single cells into a compact, biologically enriched embedding. This encoder supports enhanced cell representation through three mechanisms: (i) gene-gene relational modeling via low-rank factorization, (ii) global gene importance attention, and (iii) cell-wise aggregation using multi-head attention. It is optimized to handle variable numbers of cells per input sample, with optional masking support.

Given an input tensor of shape $[B, C_{\max}, G]$ where $B$ is the batch size, $C_{\max}$ is the maximum number of cells per patch, and $G$ is the number of genes the encoder outputs a single $[B, D_{\text{out}}]$ embedding per patch. The architecture comprises the following main components: **Gene-Gene Relation Module:** Models contextual dependencies between genes per cell using low-rank matrix factorization. **Gene Attention Module:** Applies learned global attention weights to emphasize biologically informative genes. **Cell Encoder Stack:** Processes gene expression per cell through residual blocks to obtain

latent cell embeddings. **Multi-Head Attention Aggregation:** Aggregates cell embeddings into a single patch embedding using learned cell-wise attention across multiple heads. **Final Projection and Gating:** Refines the aggregated feature through normalization, projection, and optional gating.

**Gene-Gene Relation Module:** This module enhances input gene vectors via low-rank matrix factorization:

$$\tilde{x}_i = x_i + 0.1 \cdot \left[ (x_i U_i) \, V_i \right], \quad \text{where } U_i \in \mathbb{R}^{G \times K}, V_i \in \mathbb{R}^{K \times G}$$

The $U_i$ and $V_i$ matrices are predicted per cell using a lightweight feed-forward subnetwork. The gene relation parameters $U_i \in \mathbb{R}^{G \times K}$ and $V_i \in \mathbb{R}^{K \times G}$ for each cell are generated via a two-layer MLP:

$$h_i = \text{Dropout}(\text{SiLU}(\text{LayerNorm}(W_1 x_i + b_1))) \in \mathbb{R}^{256}$$
$$\theta_i = W_2 h_i + b_2 \in \mathbb{R}^{2 \cdot G \cdot K}$$
$$\theta_i = \text{concat}(\text{vec}(U_i), \text{vec}(V_i))$$

Here $x_i \in \mathbb{R}^G$ is the input gene expression vector for cell $i$. $W_1 \in \mathbb{R}^{256 \times G}$ and $W_2 \in \mathbb{R}^{2 \cdot G \cdot K \times 256}$ are learned projection weights. $\theta_i$ is reshaped into matrices $U_i \in \mathbb{R}^{G \times K}$ and $V_i \in \mathbb{R}^{K \times G}$.

**Gene Attention Module:** A softmax-normalized attention vector $\alpha \in \mathbb{R}^G$ is learned across all genes:

$$x_i' = x_i \odot \text{softmax}(\alpha)$$

**Cell Encoder Stack:** Each weighted gene expression vector $x_i \in \mathbb{R}^G$ (or $x_i \in \mathbb{R}^{2G}$ if concatenated with auxiliary features) is passed through a stack of two residual blocks to produce a latent cell embedding $e_i \in \mathbb{R}^{256}$.

Residual Block 1:

$$h_i^{(1)} = \text{Dropout}(\text{SiLU}(\text{Linear}(\text{LayerNorm}(x_i)))) \in \mathbb{R}^{512}$$

Residual Block 2:

$$e_i = \text{Dropout}(\text{SiLU}(\text{Linear}(\text{LayerNorm}(h_i^{(1)})))) \in \mathbb{R}^{256}$$

These operations are applied independently to each cell $i$ in the sample. The resulting set $\{e_i\}_{i=1}^C$ serves as the input to the multi-head attention aggregator.

**Cell Aggregation via Multi-Head Attention:** The encoded cell embeddings $\{e_i\}_{i=1}^N$, where $e_i \in \mathbb{R}^d$, are aggregated using a multi-head attention mechanism. Each head learns to focus on different aspects of the cell population. Let $H$ be the number of attention heads. For each head $h = 1, \ldots, H$: Attention Logits compute unnormalized attention weights via a head-specific scoring network:

$$a_i^{(h)} = w^{(h)\top} \cdot \tanh(W^{(h)} e_i + b^{(h)}) \quad \text{for } i = 1, \ldots, N$$

**Attention Weights:** Normalize logits using a softmax across all $N$ cells. If masking is used to ignore padded or invalid cells, the masked version is applied:

$$\alpha_i^{(h)} = \frac{\exp(a_i^{(h)})}{\sum_{j=1}^N \exp(a_j^{(h)})} \quad \text{(no masking)}$$

$$\alpha_i^{(h)} = \frac{\exp(a_i^{(h)}) \cdot m_i}{\sum_{j=1}^N \exp(a_j^{(h)}) \cdot m_j}, \quad m_i \in \{0, 1\} \quad \text{(with masking)}$$

**Projection:** Linearly project each embedding into a head-specific space:

$$\tilde{e}_i^{(h)} = V^{(h)} e_i$$

**Aggregation:** Compute the weighted sum of projected embeddings:

$$z^{(h)} = \sum_{i=1}^N \alpha_i^{(h)} \tilde{e}_i^{(h)}$$

**Head Fusion:** The final aggregated representation is obtained by averaging over all heads:

$$z = \frac{1}{H} \sum_{h=1}^{H} z^{(h)}$$

This vector $z \in \mathbb{R}^{d'}$ serves as the RNA-level embedding summarizing the full set of cell embeddings through attention-driven aggregation.

**Final Encoding and Feature Gating:** The aggregated embedding is passed through a final linear projection and optional sigmoid gating:

$$z = \text{LayerNorm}(\text{Linear}(\text{LayerNorm}(a))), \quad \hat{z} = z \odot \sigma(\text{Linear}(z))$$

**Output:** The encoder outputs a batch of embeddings $[B, D_{\text{out}}]$, where $D_{\text{out}}$ is a configurable dimension (e.g., 512). These embeddings can be used for downstream tasks such as conditioning image synthesis via UNet.

For clarity, we summarize the main tensor shapes used:

- $x \in \mathbb{R}^{C \times H \times W}$: Feature map at each U-Net level
- $t \in \mathbb{R}^{d_t}$: Diffusion timestep embedding
- $r \in \mathbb{R}^{d_r}$: RNA embedding vector (output of RNA encoder)
- $x_i \in \mathbb{R}^G$: Raw gene expression vector for cell $i$
- $e_i \in \mathbb{R}^d$: Encoded cell embedding for cell $i$
- $z \in \mathbb{R}^{d'}$: Aggregated RNA embedding

### D.2 Conditioned U-Net Architecture with RNA and Timestep Embeddings

The conditioned U-Net integrates residual blocks with both timestep and RNA conditioning throughout its encoder, bottleneck, and decoder. This design allows the model to perform spatiotemporal image generation or transformation while incorporating gene expression context.

**Input and Conditioning Embeddings:** Let: $x_0 \in \mathbb{R}^{C \times H \times W}$ be the input image. $t \in \mathbb{R}^{d_t}$ be the diffusion timestep embedding. $r \in \mathbb{R}^{d_r}$ be the RNA embedding vector. These are passed to all residual blocks throughout the network.

**Encoder Path:** The encoder consists of $L$ levels. Each level performs One or more **ResBlocks** with conditioning:

$$x^{(l)} = \text{ResBlock}(x^{(l-1)}, t, r)$$

A **Downsampling** operation (e.g., strided convolution or pooling):

$$x_{\text{down}}^{(l)} = \text{Down}(x^{(l)})$$

Intermediate outputs are stored as **skip connections**:

$$\text{skip}^{(l)} = x^{(l)}$$

**Bottleneck:** At the lowest resolution level, additional **ResBlocks** with conditioning are applied:

$$x_{\text{bottleneck}} = \text{ResBlock}(\text{ResBlock}(x_{\text{down}}^{(L)}, t, r), t, r)$$

**Decoder Path:** The decoder also has $L$ levels and mirrors the encoder: Upsample the bottleneck or previous output:

$$x_{\text{up}}^{(l)} = \text{Up}(x^{(l+1)})$$

Concatenate with the corresponding skip connection:

$$x_{\text{cat}}^{(l)} = \text{Concat}(x_{\text{up}}^{(l)}, \text{skip}^{(l)})$$

Apply one or more **ResBlocks** with conditioning:

$$x^{(l)} = \text{ResBlock}(x_{\text{cat}}^{(l)}, t, r)$$

**Final Output Layer:**   After the final decoder level a final convolution layer maps the result to the desired number of output channels:

$$\hat{x}_0 = \text{Conv}_{\text{out}}(x^{(0)})$$

The U-Net leverages **ResBlocks with timestep and RNA conditioning** at every level to guide feature transformations based on both dynamic diffusion context ($t$) and transcriptomic content ($r$). Skip connections to preserve high-resolution spatial information across the network. Symmetric encoder-decoder structure to downsample and then upsample features, enabling efficient learning of hierarchical and context-aware representations.

**Residual Block with Timestep and RNA Conditioning:**   The input feature map $x \in \mathbb{R}^{C \times H \times W}$. A timestep embedding vector $t \in \mathbb{R}^{d_t}$. An RNA feature embedding vector $r \in \mathbb{R}^{d_r}$. Both $t$ and $r$ are projected and added to the intermediate representations of $x$ during processing.

**Input Transformations:**   The input $x$ is passed through two convolutional blocks (Conv-BN-GELU):

$$h_1 = \text{Conv}(\text{BN}(\text{GELU}_1(x)))$$
$$h_2 = \text{Conv}(\text{BN}(\text{GELU}_2(h_1)))$$

**Conditioning via Additive Projections:**   The timestep embedding $t$ and RNA embedding $r$ are each passed through separate MLPs (typically implemented as linear layers followed by non-linearities) and reshaped to be broadcastable across spatial dimensions:

$$t_{\text{proj}} = \text{MLP}_t(t) \in \mathbb{R}^C$$

$$r_{\text{proj}} = \text{MLP}_r(r) \in \mathbb{R}^C$$

These are added to $h_2$:

$$h_{\text{cond}} = h_2 + t_{\text{proj}}[:, None, None] + r_{\text{proj}}[:, None, None]$$

**Final Convolution and Residual Connection:**   The conditioned output $h_{\text{cond}}$ is passed through a final convolution:

$$h_{\text{out}} = \text{Conv}_{\text{final}}(\text{GELU}(h_{\text{cond}}))$$

A skip connection is applied. If the input and output channels differ, a $1 \times 1$ convolution (projection) is applied to $x$:

$$x_{\text{res}} = \begin{cases} x, & \text{if } C_{\text{in}} = C_{\text{out}} \\ \text{Conv}_{1 \times 1}(x), & \text{otherwise} \end{cases}$$

$$\text{Output} = x_{\text{res}} + h_{\text{out}}$$

The block allows feature maps to be conditioned on both temporal context (via timestep embedding) and transcriptomic context (via RNA embedding), enabling the model to modulate its computations dynamically based on both spatial and external biological signals. For clarity, the overall residual block transformation can be written as:

$$\text{ResBlock}(x, t, r) = \text{Conv}_{\text{final}}(\text{GELU}(\text{Conv}_2(\text{GELU}(\text{Conv}_1(x))) + \text{MLP}_t(t) + \text{MLP}_r(r))) + x_{\text{res}}$$

# E   Spatial Regularization

## E.1   Spatial Graph Loss Framework

Let $\mathbf{X}_{real} \in \mathbb{R}^{B \times C \times H \times W}$ and $\mathbf{X}_{gen} \in \mathbb{R}^{B \times C \times H \times W}$ denote batches of real and generated images, where $B$ is the batch size, $C$ is the number of channels, and $H \times W$ is the spatial resolution. For each image pair in the batch, we construct a spatial graph $\mathcal{G} = (\mathcal{V}, \mathcal{E})$ where vertices $\mathcal{V}$ correspond to spatial locations and edges $\mathcal{E}$ connect spatially proximate regions.

The overall training objective combines the base rectified flow loss with the spatial graph loss:

$$\mathcal{L}_{total} = \mathcal{L}_{RF}(\mathbf{v}_\theta, \mathbf{v}_{target}) + \lambda_s \cdot w(t) \cdot \mathcal{L}_{spatial}(\mathbf{X}_{gen}, \mathbf{X}_{real}) \tag{11}$$

where $\mathcal{L}_{RF}$ is the rectified flow velocity matching loss, $\lambda_s$ is the spatial loss weight, $w(t)$ is a warmup schedule, and $\mathcal{L}_{spatial}$ enforces spatial consistency.

The warmup schedule is defined as:

$$w(t) = \begin{cases} 0 & t < t_{start} \\ \frac{t-t_{start}}{t_{warmup}} & t_{start} \leq t < t_{start} + t_{warmup} \\ 1 & t \geq t_{start} + t_{warmup} \end{cases} \quad (12)$$

where $t$ is the current epoch, $t_{start}$ is the epoch to begin spatial loss, and $t_{warmup}$ is the number of warmup epochs.

## E.2 Segmentation-Based Spatial Loss

The segmentation-based approach explicitly models cell nuclei through instance segmentation, enabling fine-grained morphological analysis.

### E.2.1 Nuclear Segmentation

We apply a pretrained Cellpose model to segment individual nuclei. For an image $\mathbf{X}$, Cellpose produces an instance segmentation mask $\mathbf{M} \in \mathbb{Z}^{H \times W}$ where $\mathbf{M}(i,j) = n$ indicates pixel $(i,j)$ belongs to nucleus $n$.

The set of detected nuclei is:

$$\mathcal{N} = \{n_1, n_2, \ldots, n_K\} \quad (13)$$

For each nucleus $n \in \mathcal{N}$, we extract its centroid:

$$\mathbf{c}_n = (\bar{i}_n, \bar{j}_n) = \left( \frac{1}{|\mathcal{R}_n|} \sum_{(i,j) \in \mathcal{R}_n} i, \frac{1}{|\mathcal{R}_n|} \sum_{(i,j) \in \mathcal{R}_n} j \right) \quad (14)$$

where $\mathcal{R}_n = \{(i,j) : \mathbf{M}(i,j) = n\}$ is the region of nucleus $n$.

### E.2.2 Morphological Feature Extraction

For each nucleus, we compute morphological features:

1. **Area:** $A_n = |\mathcal{R}_n|$
2. **Perimeter:** $P_n = \sum_{(i,j) \in \mathcal{R}_n} \mathbb{K}[\exists(i',j') \in \mathcal{N}_8(i,j) : \mathbf{M}(i',j') \neq n]$
3. **Circularity:** $C_n = \frac{4\pi A_n}{P_n^2}$
4. **Eccentricity:** Computed from the eigenvalues $\lambda_1 \geq \lambda_2$ of the covariance matrix of pixel positions:

$$E_n = \sqrt{1 - \frac{\lambda_2}{\lambda_1}} \quad (15)$$

5. **Solidity:** $S_n = \frac{A_n}{A_{convex\_hull}(n)}$

The morphological feature vector for nucleus $n$ is:

$$\mathbf{f}_{morph}(n) = [A_n, P_n, C_n, E_n, S_n] \quad (16)$$

### E.2.3 Nuclear Spatial Graph

We construct a k-nearest neighbor graph in the space of nuclear centroids:

$$\mathcal{G}_{nuclei} = (\mathcal{N}, \mathcal{E}_{nuclei}) \tag{17}$$

where edges connect spatially proximate nuclei:

$$\mathcal{E}_{nuclei} = \{(n_i, n_j) : j \in \text{top-}k \text{ nearest neighbors of } i\} \tag{18}$$

based on centroid distances $d(\mathbf{c}_{n_i}, \mathbf{c}_{n_j}) = \|\mathbf{c}_{n_i} - \mathbf{c}_{n_j}\|_2$.

### E.2.4 Morphological Consistency Loss

For matched nuclei between real and generated images (matched by spatial proximity of centroids), we penalize morphological differences:

$$\mathcal{L}_{morph} = \frac{1}{|\mathcal{N}_{matched}|} \sum_{n \in \mathcal{N}_{matched}} \|\mathbf{f}_{morph}^{real}(n) - \mathbf{f}_{morph}^{gen}(n)\|_2 \tag{19}$$

where $\mathcal{N}_{matched}$ is the set of matched nuclei pairs.

### E.2.5 Spatial Arrangement Loss

We enforce consistency in the spatial arrangement of neighboring nuclei. For each nucleus $n_i$ and its neighbors $\mathcal{N}_k(n_i)$:

$$\mathcal{L}_{arrangement} = \frac{1}{|\mathcal{N}|} \sum_{n_i \in \mathcal{N}} \frac{1}{k} \sum_{n_j \in \mathcal{N}_k(n_i)} \left| d_{real}(\mathbf{c}_{n_i}, \mathbf{c}_{n_j}) - d_{gen}(\mathbf{c}_{n_i}, \mathbf{c}_{n_j}) \right| \tag{20}$$

This term encourages similar inter-nuclear distances in generated images.

### E.2.6 Nuclear Density Consistency

We compare local nuclear density using kernel density estimation. The nuclear density at location $(i, j)$ is:

$$\rho(i, j) = \sum_{n \in \mathcal{N}} \mathcal{K}_h(\|\mathbf{c}_n - (i, j)\|_2) \tag{21}$$

where $\mathcal{K}_h$ is a Gaussian kernel with bandwidth $h$:

$$\mathcal{K}_h(d) = \frac{1}{\sqrt{2\pi h^2}} \exp\left(-\frac{d^2}{2h^2}\right) \tag{22}$$

The density consistency loss is:

$$\mathcal{L}_{density} = \frac{1}{HW} \sum_{i=1}^{H} \sum_{j=1}^{W} |\rho_{real}(i, j) - \rho_{gen}(i, j)| \tag{23}$$

### E.2.7 Combined Segmentation-Based Loss

$$\mathcal{L}_{spatial}^{segment} = \beta_{morph} \cdot \mathcal{L}_{morph} + \beta_{arr} \cdot \mathcal{L}_{arrangement} + \beta_{dens} \cdot \mathcal{L}_{density} \tag{24}$$

where $\beta_{morph}, \beta_{arr}, \beta_{dens}$ are weighting hyperparameters.

### E.3 Gradient-Based Spatial Loss

The gradient-based approach captures local texture patterns through image derivatives and neighborhood similarity. For each spatial location $(i, j)$, we extract a local patch and compute its features.

#### E.3.1 Gradient Feature Extraction

We compute spatial gradients using Sobel operators:

$$\mathbf{G}_x = \mathbf{X} * \mathbf{K}_x, \quad \mathbf{G}_y = \mathbf{X} * \mathbf{K}_y \tag{25}$$

where $*$ denotes convolution and $\mathbf{K}_x, \mathbf{K}_y$ are Sobel kernels:

$$\mathbf{K}_x = \begin{bmatrix} -1 & 0 & 1 \\ -2 & 0 & 2 \\ -1 & 0 & 1 \end{bmatrix}, \quad \mathbf{K}_y = \begin{bmatrix} -1 & -2 & -1 \\ 0 & 0 & 0 \\ 1 & 2 & 1 \end{bmatrix} \tag{26}$$

The gradient magnitude and orientation at each pixel are:

$$M(i, j) = \sqrt{G_x(i, j)^2 + G_y(i, j)^2}, \quad \theta(i, j) = \arctan\left(\frac{G_y(i, j)}{G_x(i, j)}\right) \tag{27}$$

#### E.3.2 Texture Feature Extraction

We extract local texture features using patch statistics. For a patch centered at $(i, j)$ with radius $r$:

$$\mathcal{P}_{i,j} = \{\mathbf{X}(i', j') : |i' - i| \leq r, |j' - j| \leq r\} \tag{28}$$

The texture feature vector $\mathbf{f}_{texture}(i, j)$ includes:

$$\mathbf{f}_{texture}(i, j) = [\mu(\mathcal{P}_{i,j}), \sigma(\mathcal{P}_{i,j}), s(\mathcal{P}_{i,j}), k(\mathcal{P}_{i,j})] \tag{29}$$

where $\mu, \sigma, s, k$ are the mean, standard deviation, skewness, and kurtosis of the patch.

#### E.3.3 Spatial Graph Construction

We construct a k-nearest neighbor graph in spatial coordinates. For a downsampled grid of locations $\{(i_1, j_1), \ldots, (i_N, j_N)\}$, we find the $k$ nearest neighbors for each location based on Euclidean distance:

$$\mathcal{N}_k(i_m, j_m) = \{(i_n, j_n) : d((i_m, j_m), (i_n, j_n)) \in \text{top-}k \text{ smallest}\} \tag{30}$$

where $d((i_m, j_m), (i_n, j_n)) = \sqrt{(i_m - i_n)^2 + (j_m - j_n)^2}$.

#### E.3.4 Gradient-Based Spatial Loss

The gradient component of the spatial loss compares gradient patterns between spatially neighboring locations:

$$\mathcal{L}_{gradient} = \frac{1}{N} \sum_{m=1}^{N} \frac{1}{k} \sum_{(i_n, j_n) \in \mathcal{N}_k(i_m, j_m)} \|\mathbf{G}_{real}(i_m, j_m) - \mathbf{G}_{gen}(i_m, j_m) - (\mathbf{G}_{real}(i_n, j_n) - \mathbf{G}_{gen}(i_n, j_n))\|_2 \tag{31}$$

where $\mathbf{G} = [G_x, G_y]$ is the gradient vector. This formulation enforces that gradient differences between neighbors should be similar in real and generated images.

### E.3.5 Texture-Based Spatial Loss

The texture component compares local texture statistics:

$$\mathcal{L}_{texture} = \frac{1}{N} \sum_{m=1}^{N} \frac{1}{k} \sum_{(i_n,j_n) \in \mathcal{N}_k(i_m,j_m)} \|\mathbf{f}_{texture}^{real}(i_m,j_m) - \mathbf{f}_{texture}^{gen}(i_m,j_m)\|_2 \qquad (32)$$

### E.3.6 Combined Simple Spatial Loss

$$\mathcal{L}_{spatial}^{simple} = \alpha_{grad} \cdot \mathcal{L}_{gradient} + \alpha_{tex} \cdot \mathcal{L}_{texture} \qquad (33)$$

where $\alpha_{grad}$ and $\alpha_{tex}$ are weighting factors (typically $\alpha_{grad} = 1.0, \alpha_{tex} = 0.5$).

### E.4 Implementation Details

Both methods use:

- **k-nearest neighbors:** $k = 5$
- **Spatial loss weight:** $\lambda_s = 0.1$
- **Warmup epochs:** 5 epochs with linear ramp-up
- **Activation threshold:** Spatial loss begins at 70% of total epochs or when validation loss drops below a predetermined threshold

The gradient-based method is computed at every training step with negligible overhead ($\sim$5% increase in training time), while the segmentation-based method uses cached segmentation masks updated every few epochs to balance accuracy and computational cost.

## F  Data Availability and License

We used the following 10x Xenium demo data:

- **Dataset 1 (Xenium $C1$):**
  Human Skin Preview Data (Xenium Human Skin Gene Expression Panel), In Situ Gene Expression dataset analyzed using Xenium Onboard Analysis 1.6.0, 10x Genomics, (2023-09-19).

- **Dataset 2 (Xenium $C2$):**
  Human Skin Preview Data (Xenium Human Skin Gene Expression Panel with Custom Add-On), In Situ Gene Expression dataset analyzed using Xenium Onboard Analysis 1.7.0, 10x Genomics, (2023-12-08).

- **Dataset 3 (Xenium $P1$):**
  Preview Data: FFPE Human Skin Primary Dermal Melanoma with 5K Human Pan Tissue and Pathways Panel, In Situ Gene Expression dataset analyzed using Xenium Onboard Analysis 3.0.0, 10x Genomics, (2024-08-01).

These datasets are licensed under the Creative Commons Attribution 4.0 International (CC BY 4.0) license, as indicated in the dataset documentation.

**HEST-1k:** The dataset is public available on HuggingFace. The dataset is distributed under the Attribution-NonCommercial-ShareAlike 4.0 International license (CC BY-NC-SA 4.0 Deed).

