# OpenReview forum: "GeneFlow: Translation of Single-cell Gene Expression to Histopathological Images via Rectified Flow"
_NeurIPS.cc/2025/Conference — NeurIPS 2025 poster_

### Official Review · Reviewer_Uins · 2025-06-27

**Clarity:** 3
**Significance:** 4
**Originality:** 3
**Rating:** 5
**Confidence:** 4

**Summary:**

This paper introduces GeneFlow, a generative model that synthesizes histopathological images (e.g., H&E, DAPI) from single-cell or multi-cell gene expression data, leveraging a rectified flow framework conditioned on expression-derived embeddings. The architecture integrates a gene encoder that utilizes low-rank gene-gene interactions, attention mechanisms, and multi-cell aggregation into a UNet, which predicts velocity fields under rectified flow dynamics. The method is evaluated on three 10x Xenium melanoma datasets, demonstrating substantial improvements over diffusion-based baselines across FID, SSIM, and expert preference scores. The authors further conduct gene importance analysis via gradient-based sensitivity and demonstrate meaningful enrichment in known cancer-related pathways. Overall, the paper addresses an underexplored direction—translating gene expression into visual morphology—using a principled generative framework and delivers promising empirical results.

**Questions:**

1. Spatial encoding: Why are cell coordinates ignored? Given that histology reflects spatial organization, and spatial transcriptomics provides this information, it is surprising that no (x, y) information is used. Incorporating spatial proximity (e.g., via GNNs, positional encoding, or coordinate-aware attention) would likely improve biological realism.
2. Single-cell image generation: How biologically meaningful is it to generate a 256×256 histological image from a single cell’s gene expression? What downstream use cases motivate this, and could the image instead reflect cell morphology rather than tissue context?
3. Rectified flow vs. alternatives: Can the authors provide ablations comparing rectified flow to direct conditional generation (e.g., UNet decoding without flow)? What are the practical benefits of rectified flow in this setting beyond improved FID—e.g., smoother interpolation, faster convergence, or better generalization?
4. Biological validation: Have the authors considered validating biological structure in generated images? For example:
- Use cell type classifiers to test if generated images preserve type-specific morphology.
- Compare nuclear features (size, shape) between real and generated images.
- Evaluate tissue-level arrangements (e.g., immune niches, epithelial boundaries).

**Ethical Concerns:**

["NO or VERY MINOR ethics concerns only"]

**Final Justification:**

I appreciate the author's effort to address my concerns and improve the quality of the paper. The authors have successfully clarified my concerns, and I will increase the rate by one score (from 4 to 5).

**Limitations:**

Same as the questions.

**Quality:**

3

**Strengths And Weaknesses:**

Strengths
1. Novel problem formulation: The paper tackles the relatively underexplored inverse mapping from gene expression to histology, complementing prior work that predicts expression from images.
2. Sound architectural design: The integration of low-rank gene interaction modeling, global attention, and multi-head cell aggregation into a rectified flow UNet is well-motivated.
3. Strong empirical performance: The method consistently outperforms diffusion baselines with large improvements in FID (e.g., 213 $\rightarrow$ 34), and is preferred by expert pathologists in 86% of blind tests.
4. Cross-dataset generalization: Despite differing gene panels, the model shows reasonable robustness when transferring between datasets (e.g., C1 $\rightarrow$ P1).

Weaknesses
1. No modeling of spatial coordinates: The most critical limitation is that the model entirely ignores spatial (x, y) coordinates of cells, despite operating on spatial transcriptomics data. Cells are treated as unordered sets, which breaks the spatial premise of histology and undermines biological plausibility.
2. Ambiguous use of single-cell vs. multi-cell modes: The distinction between these modes is unclear, and generating full image patches from single cells is biologically questionable. The architecture is nearly identical between the two modes.
3. Overreliance on perceptual metrics: FID and SSIM are used to claim biological realism, but no explicit evaluation of biological structure (e.g., tissue architecture, morphometric consistency) is provided.
4. Lack of ablation for architectural necessity: The benefits of rectified flow (e.g., invertibility, smooth interpolation, or data efficiency) are asserted but not empirically demonstrated. It’s unclear whether simpler models (e.g., conditional UNet with direct regression) could achieve similar results.
5. Limitations underdeveloped: The Discussion section does not explicitly acknowledge the absence of spatial modeling, which is arguably the most critical shortcoming of the current implementation.

---

> ### Author Rebuttal · Authors · 2025-07-30
>
> We thank the reviewer for their valuable feedback. The questions raised were extremely important for a more comprehensive evaluation and validation of our method. We have answered every question, performed more experiments, and explained the absence of spatial modelling which was the top concern by the reviewer. The complete set of results on all datasets could not fit the character limit. Those along with their corresponding images will be available in the final version or provided during the discussion period. Please let us know if further explanation or experiments needed to improve the quality of the final version of the paper.
>
> > Distinction between single-cell and multi-cell models is unclear, and generating full patches from single cells seems biologically questionable?
>
> We acknowledge the potential confusion and will clarify this in the final version. The single-cell model generates 256×256 patches centered on a focal cell, using that cell’s gene expression as the conditioning input while preserving surrounding tissue context for realistic morphology, with preferential weighting over layout. In contrast, the multi-cell model aggregates gene expression from all cells in a patch using multi-head attention, capturing collective molecular states. While the core UNet architecture is shared, the RNA encoders differ: direct input for single-cell versus aggregated attention for multi-cell. Single-cell mode reflects how an individual cell’s transcriptome influences its immediate microenvironment, enabling study of cell-autonomous effects, whereas multi-cell mode models more complex intercellular interactions and heterogeneity. Training data preparation differs accordingly: single-cell patches are extracted centered on valid cells, yielding more homogeneous data enriched in melanocytes, whereas multi-cell patches use sliding windows over heterogeneous tissue. This distinction, alongside dataset biases, explains why single-cell models tend to perform better and produce purer cell-type reconstructions.
>
> > Evaluation relies too heavily on perceptual metrics like FID and SSIM; biological structural validation is missing?
>
> Although FID uses Inception features trained on natural images, it provides useful relative benchmarks of image quality within our domain. We further explored emerging histopathology-specific evaluation metrics using UNI2-h foundational model:
>
> UNI2-h FID: Domain-specific image quality assessment using a pathology foundation model.
>
> UNI2-h embedding similarity: Comparing feature distributions of real and generated images.
>
> Nuclear morphometric similarity: Quantification of circularity, eccentricity, and solidity from segmented nuclei, statistically compared between real/generated images.
>
> Spatial energy similarity: Gray-level co-occurrence matrix energy.
>
> Spatial complexity and feature magnitude: Derived from UNI2-h embeddings, capturing tissue structure.
>
> Calculated these metrics across P1 C1 and C2 datasets but due to character constraints only presented C1 dataset comparison in the table below. Future work will integrate these metrics to better illustrate and diagnose specific multi-cell model limitations such as boundary definition and texture fidelity.
>
> ### Table: Evaluation Metrics Comparison for C1 Dataset: Multi-cell and Single-cell Models
> || Metric| C1 Multi Diff.| C1 Multi Rect.| C1 Single Diff.| C1 Single Rect.|
> |-----------------------|--------------------------------|---------------------|---------------------|----------------------|----------------------|
> | **Image Quality Metrics ↓** ||||||
> || FID Overall UNI2-h| 404.06| **87.96**| 405.98 | **39.27**|
> | **Biological Feature Similarity ↑** | |||||
> || UNI2-h Embedding Sim.   | 0.967±0.007  | **0.979±0.004**| 0.969±0.007   | **0.983±0.003**|
> || Nuclear Circularity Sim.| 0.835±0.043  | **0.844±0.037**| 0.839±0.049   | **0.874±0.028**|
> || Nuclear Eccentricity Sim.| 0.869±0.046  | **0.954±0.015**| 0.880±0.039   | **0.964±0.012**|
> || Nuclear Solidity Sim.   | 0.714±0.048  | **0.888±0.036**| 0.721±0.077   | **0.867±0.028**|
> | **Spatial Feature Metrics ↑** ||||||
> || Spatial Energy Sim.| -| **0.283±0.083**| -| **0.678±0.176**|
> || Spatial Complexity Sim. | 0.167±0.138  | **0.506±0.080**| 0.130±0.111   | **0.571±0.085**|
> || Spatial Feat. Magnitude Sim.   | 0.167±0.143  | **0.514±0.077**| 0.135±0.119   | **0.571±0.084**|
>
> > Benefits of rectified flow over simpler models are asserted but not empirically demonstrated; request ablation versus conditional UNet regression?
>
> Rectified flow models inherently handle many-to-one gene-to-morphology mappings better than deterministic regressors, as supported by prior literature showing their simpler training and more efficient sampling compared to diffusion and autoencoders.
>
> Per reviewer suggestion, we implemented a conditional UNet baseline with our gene encoder trained on direct regression losses (MSE, perceptual), results detailed in the included ablation table. The transformer-based RNA encoder outperforms simpler variants across all measured metrics (Section C2, Figure 7), and each encoder module is critical for interpretability (Section 4.3, Figure 4);  Removing any module reduces gene importance detection and biological insight. These results justify the encoder’s complexity: slight performance trade-offs enable enhanced interpretability essential for understanding gene-morphology relationships.
>
> We also extended experiments across the 59-sample HEST-1k dataset (12 organs, 1.6 million patches) to demonstrate scalability and cross-tissue generalization, with promising preliminary single-cell training underway. Cross-dataset tests highlight strong generalization despite limited gene overlap (~126/5000 shared genes), supporting that our model learns generalizable gene-morphology mappings rather than dataset-specific artifacts. Supplementary visuals corroborate these results.
>
> ### Table: C1 Single-Cell Ablation Studies and Baseline | HEST-1k Dataset Multi-Cell
> || Metric| UNet (MSE)| Diffusion | Rectified | -Gene Att.| -Gene Att./Rel.| -Gene Att./Multi Att. | HEST-1k|
> |-----------------------|-------------------------------|-------------------|--------------------|--------------------|--------------------|---------------------|------------------------|---------------------|
> | **Image Quality Metrics ↓** || |||||||
> || FID Overall UNI2-h| 525.51| 405.98| 39.27| **26.46** | 27.25| 29.50| 62.54|
> || Inception Feat. Dist.| 21.76±1.29| 20.18±1.61| **14.50±2.52**| 14.89±2.52| 15.02±2.70 | 15.39±2.68| 15.26±2.70 |
> | **Biological Feature Similarity ↑** || |||||||
> || UNI2-h Embedding Sim.| 0.964±0.004 | 0.969±0.007| 0.983±0.003| **0.991±0.003**| 0.990±0.003| 0.990±0.003| 0.974±0.004|
> || Nuclear Circularity Sim.| 0.659±0.038 | 0.839±0.049| 0.874±0.028| **0.949±0.022**| 0.941±0.021| 0.947±0.020| 0.924±0.027|
> || Nuclear Eccentricity Sim.| 0.655±0.024 | 0.880±0.039| 0.964±0.012| **0.959±0.013**| 0.954±0.015| 0.957±0.017| 0.955±0.014|
> || Nuclear Solidity Sim.| 0.479±0.037 | 0.721±0.077| 0.867±0.028| **0.950±0.020**| 0.943±0.019| 0.949±0.020| 0.921±0.028|
> | **Spatial Feature Metrics ↑** || |||||||
> || Spatial Energy Sim.| 0.018±0.034 | -| 0.678±0.176| **0.723±0.130**| 0.735±0.055| 0.719±0.048| 0.744±0.049|
> || Spatial Complexity Sim. | 0.099±0.060 | 0.130±0.111| 0.571±0.085| **0.749±0.069**| 0.704±0.072| 0.721±0.067| 0.641±0.061|
> || Spatial Feat. Magnitude Sim.| 0.112±0.061 | 0.135±0.119| 0.571±0.084| **0.759±0.068**| 0.718±0.068| 0.735±0.069| 0.635±0.059|
>
>
> > The model ignores spatial (x, y) coordinates of cells; this omission weakens biological plausibility and limits spatial modeling capacity?
>
> We acknowledge this limitation and will clarify in the final version. Our approach prioritizes gene expression as the main determinant of morphology at the patch level, focusing on local tissue microenvironments where relative cellular positioning and molecular context suffice. Patch sizes (256×256 pixels) typically contain fewer than 50 cells with limited heterogeneity, reducing the necessity for explicit global coordinate modeling. Spatial coordinates are slide-specific, noisy, and vary across experiments, limiting their generalizability, while relative spatial relationships are inherently encoded in histology images and indirectly inferred by our model. Our preliminary attempts to incorporate explicit coordinate recovery degraded performance, highlighting the need for more sophisticated, invariant geometric representations. We are actively developing hierarchical frameworks integrating spatially aware tile generation and multi-scale reasoning to model both local and global tissue organization. Computational complexity demands careful architectural design; thus, we focused on this critical foundational step first.
>
> > Biologically meaningful generating a 256×256 histological image from single-cell gene expression? Downstream Application?
>
> The single-cell generation serves as a proof-of-concept to isolate how individual transcriptomes influence cellular morphology and microenvironmental context, providing insights complementary to the multi-cell model’s tissue-level aggregation. This dual modality enables applications in drug mechanism studies, functional genomics, and biomarker discovery, where cross-modal prediction has demonstrated impact. Single-cell morphology prediction supports virtual cytomorphology for diagnosis and reveals how cell-intrinsic and extrinsic factors shape phenotypes, important in immunology and oncology for understanding paracrine signaling and tumor-immune interactions. Pathologist collaborators confirmed that even 256×256 images centered on a single cell contain morphological cues sufficient to broadly classify tumor categories, demonstrating biological relevance. This approach facilitates mechanistic insight and hypothesis testing challenging to obtain from bulk or patch-level analyses alone.

---

> > ### Comment · Reviewer_Uins · 2025-08-04
> >
> > I appreciate the author's effort to address my concerns and improve the quality of the paper. The authors have successfully clarified my concerns, and I will increase the rate by one score (from 4 to 5).

---

> > > ### Author Response · Authors · 2025-08-05
> > > **Gentle Reminder**
> > >
> > > Dear Reviewer Uins,
> > >
> > > Thank you again for your thoughtful feedback and your intention to raise the score from 4 to 5.
> > >
> > > We have noticed that change in the score has not reflected on Open Review for our submission. This is a gentle reminder to please update the rating whenever you find the time before the discussion period ends.
> > >
> > > Best Regards,
> > >
> > > Authors

---

> ### Author Response · Authors · 2025-08-04
>
> Thank you!
>
> We are deeply grateful to Reviewer Uins for their thoughtful engagement and positive assessment of our work. We sincerely appreciate the time they invested in our manuscript and are especially thankful for intending to raise the score from 4 to 5.
>
> Their excellent suggestions are invaluable, and we look forward to incorporating them to enhance our method's thoroughness  in the final version. Should the reviewer have any additional questions, we would be delighted to address them promptly.

---

### Official Review · Reviewer_Av8N · 2025-07-02

**Clarity:** 3
**Significance:** 2
**Originality:** 2
**Rating:** 4
**Confidence:** 3

**Summary:**

This paper introduces GeneFlow, a method that translates single and multi-cell gene expressions into histopathological images. GeneFlow uses an attention-based RNA encoder to condition a rectified flow model trained to generate histopathological images. The method can be applied to both single-cell and multi-cell cases. It is compared against variants using diffusion models instead of rectified flow, to show improved generation quality.

**Questions:**

1. Can the authors better articulate the novelty of the method beyond the architectural design of the RNA encoder?

2. The paper proposes to tackle the inverse process (transcriptomics data to histopathological images). Given the difficulty in finding baselines for this task, the authors should clearly demonstrate the utility of addressing such a problem. Additionally, ablation studies on different design choices are needed. While the generative component has been compared between diffusion models and rectified flow, the other components should also be validated. For example, can the authors provide a discussion or ablation study on the impact of the gene encoder? Why focus on the raw gene count data instead of leveraging pretrained models such as scBERT (Yang et al., 2022)? he main contribution seems to lie in the gene encoder design, it is important to evaluate its performance by considering alternative embeddings.

3. Can the authors provide ilmproved evaluation measures? For example, a cross-modal coherence metric could be used since models exist that translate histopathological images to RNA transcriptomes. Such metrics would help assess whether the mapping has been effectively learned. State-of-the-art models for histopathological image to RNA transcriptome translation can be used for this evaluation.
4. Since FID is based on the Inception model trained on natural images, could the authors consider alternative classifiers trained on histopathological images to obtain more reliable evaluation?

5. Comments on presentation:
- The best performance results should be highlighted in bold.
- The interpretation of evaluation metrics (higher or lower is better) should be clearly indicated in the tables.

**Reference:**
Yang, F., Wang, W., Wang, F., Fang, Y., Tang, D., Huang, J., ... & Yao, J. (2022). scBERT as a large-scale pretrained deep language model for cell type annotation of single-cell RNA-seq data. *Nature Machine Intelligence*, 4(10), 852-866.

**Ethical Concerns:**

["NO or VERY MINOR ethics concerns only"]

**Final Justification:**

The paper presents a methodological study applying generative models to the biological domain, which is important and has the potential to benefit both practitioners and researchers. The authors have conducted new experiments and incorporated evaluation metrics that are better suited to the application, effectively addressing my previous concerns. Therefore, I have raised my score to 4 and recommend acceptance.

**Limitations:**

Yes.

**Paper Formatting Concerns:**

No issue, noticed

**Quality:**

2

**Strengths And Weaknesses:**

**Strengths**

1. The paper presents a methodological work which applies generative models to the biological domain, which is important and can help practitioners and researchers.
2. The paper is clear and easy to follow.

**Weaknesses**

1. My main concern is with the empirical validation which could be significantly improved. The evaluation metric used FID may not be suitable, as the Inception model was trained on ImageNet and may not generalize well to histopathological data. More appropriate evaluation metrics should be considered.
2. The paper assumes that paired single-cell RNA and corresponding images are always available, which is a strong assumption in this domain.
3. The practical utility of solving the inverse problem (gene expression to images) is unclear, as most existing work focuses on the other way around. It would be helpful to clarify this and discuss wether the method can also be applied in this direction as well.

---

> ### Author Rebuttal · Authors · 2025-07-30
>
> We thank the reviewer for their valuable feedback. The critiques mentioned were extremely fair and important for a more comprehensive evaluation and validation of our method. We have answered every question regarding novelty, performed more experiments with a new larger dataset, and explored biologically relevant metrics which was the top concern by the reviewer. The complete set of results on all datasets could not fit the character limit. Those along with new images will be available in the final version or provided during the discussion period. Please let us know if further explanation or experiments needed to improve the quality of the final version of the paper. With this, we hope the reviewer recognizes the quality, significance and originality of our research and consider to raise the score.
>
> > Paper assumes paired single-cell RNA images are always available?
>
> Paired single-cell RNA and histology images are scarce due to technological constraints. We address this by using 10x Xenium datasets offering true subcellular transcriptomic profiles co-registered with images, enabling our framework to learn morphology from real paired data. Our method also supports near sc-level ST data with transcriptomics of mixed cells, leveraging spot cellularity modeling. We further trained and evaluated multi-cell model on 59 Xenium samples spanning 12 organs (HEST-1k dataset, 1.6 million patches post reprocessing), showing generalizability beyond melanoma and across tissues, with single-cell model results forthcoming due to longer training. The aim of our paper is a solution to this very problem.
>
> >  Practical utility of solving the inverse problem?
>
> Inverse model unlocks applications beyond morphology recovery for scRNA-seq data lacking spatial context. 1. Enables predictive visualization of morphological changes under gene perturbations (gene knockdown, drug effects), facilitating clinical diagnosis, in silico hypothesis generation and drug repurposing screens. 2. Gene importance analyses identify driver genes within clinically relevant pathways such as EMT and angiogenesis, supporting biomarker discovery. 3. Quality control tool in ST by flagging discrepancies between predicted and true histology that may signal artifacts or tissue issues. Ongoing collaborations with pathologists aim to validate these utilities. While existing work focuses on forward prediction from images to gene expression, our inverse framework complements and expands these research avenues with generative tissue modeling.
>
> > Novelty beyond the RNA encoder?
>
> Novelty extends well beyond encoder design, four key contributions: 1. Pioneering ST inverse mapping with rectified flow dynamics, outperforming baselines; 2. Multi-head attention module for aggregating variable numbers of cells in spatial patches; 3. Gene relation network using low-rank factorization for interpretable, efficient gene-gene interaction modeling. This allows the model to learn both single and multi cell level information.; 4. Biological validation via pathway enrichment showing the model autonomously identifies clinically relevant signals.
>
> Collectively, these innovations establish the first comprehensive framework tackling this inverse problem and set a new research direction linking generative modeling with spatial biology. The problem we solve is the most novel aspect of this research. Proven by the baseline comparisons which could not faithfully generate high quality cell/tissue level images.
>
> > Raw gene counts instead of pretrained models like scBERT?
>
> Raw gene counts over pretrained embeddings like scBERT due to the specific requirements of ST data and multi-cell patch modeling. scBERT excels in single-cell embeddings for classification but lacks mechanisms for multi-cell spatial aggregation and gene-gene relational modeling critical to our image generation task. scBERT was trained on scRNA-seq with thousands of genes. Our ST datasets comprise smaller, variable targeted panels with limited gene overlap, reducing scBERT’s applicability. Our custom encoder preserves quantitative gene-level precision and interpretable gene relations tailored to morphology prediction, which pretrained embeddings designed for cell state representation do not capture.
>
> > Ablations on gene encoder?
>
> We implemented and compared a conditional UNet baseline using our gene encoder trained with MSE and perceptual losses, confirming rectified flow’s advantages in capturing complex gene-to-image mappings. Our ablation study (Section C2, Figure 7) shows the transformer-based RNA encoder consistently outperforms simpler alternatives across all metrics, with each component critical for interpretability analyses (Section 4.3, Figure 4). Removing any module reduces gene importance detection and biological insight. These results justify the encoder’s complexity: slight performance trade-offs enable enhanced interpretability essential for understanding gene-morphology relationships.
>
> ### Table: C1 Single-Cell Ablation Studies and Baseline | HEST-1k Dataset Multi-Cell
> || Metric| UNet (MSE)| Diffusion | Rectified | -Gene Att.| -Gene Att./Rel.| -Gene Att./Multi Att. | HEST-1k|
> |-----------------------|-------------------------------|-------------------|--------------------|--------------------|--------------------|---------------------|------------------------|---------------------|
> | **Image Quality Metrics ↓** || |||||||
> || FID Overall UNI2-h| 525.51| 405.98| 39.27| **26.46** | 27.25| 29.50| 62.54|
> || Inception Feat. Dist.| 21.76±1.29| 20.18±1.61| **14.50±2.52**| 14.89±2.52| 15.02±2.70 | 15.39±2.68| 15.26±2.70 |
> | **Biological Feature Similarity ↑** || |||||||
> || UNI2-h Embedding Sim.| 0.964±0.004 | 0.969±0.007| 0.983±0.003| **0.991±0.003**| 0.990±0.003| 0.990±0.003| 0.974±0.004|
> || Nuclear Circularity Sim.| 0.659±0.038 | 0.839±0.049| 0.874±0.028| **0.949±0.022**| 0.941±0.021| 0.947±0.020| 0.924±0.027|
> || Nuclear Eccentricity Sim.| 0.655±0.024 | 0.880±0.039| 0.964±0.012| **0.959±0.013**| 0.954±0.015| 0.957±0.017| 0.955±0.014|
> || Nuclear Solidity Sim.| 0.479±0.037 | 0.721±0.077| 0.867±0.028| **0.950±0.020**| 0.943±0.019| 0.949±0.020| 0.921±0.028|
> | **Spatial Feature Metrics ↑** || |||||||
> || Spatial Energy Sim.| 0.018±0.034 | -| 0.678±0.176| **0.723±0.130**| 0.735±0.055| 0.719±0.048| 0.744±0.049|
> || Spatial Complexity Sim. | 0.099±0.060 | 0.130±0.111| 0.571±0.085| **0.749±0.069**| 0.704±0.072| 0.721±0.067| 0.641±0.061|
> || Spatial Feat. Magnitude Sim.| 0.112±0.061 | 0.135±0.119| 0.571±0.084| **0.759±0.068**| 0.718±0.068| 0.735±0.069| 0.635±0.059|
>
> > Improve evaluation using cross-modal metrics?
>
> Evaluated cross-modal coherence by processing both ground-truth and generated images through the HE2RNA model (Schmauch et al., 2020) to predict RNA expression and computed Pearson correlations against actual RNA profiles. High correlations observed for validation sets: 0.991 ± 0.002 (C1) and 0.986 ± 0.001 (C2), indicating strong coherence between generated histology and transcriptomes. However, HE2RNA’s limited sensitivity to subtle histological variations suggests future improvements in forward models are vital for more discriminative evaluation. We will include these results in the final version.
>
> > FID on a natural image model, histopathology-specific evaluation metrics?
>
> While FID leverages Inception features trained on natural images, it remains useful for relative comparisons within histology domains by capturing coarse texture and structural distributions. We further explored emerging histopathology-specific evaluation metrics using UNI2-h foundational model:
>
> UNI2-h FID: Domain-specific image quality assessment using a pathology foundation model.
>
> UNI2-h embedding similarity: Comparing feature distributions of real and generated images.
>
> Nuclear morphometric similarity: Quantification of circularity, eccentricity, and solidity from segmented nuclei, statistically compared between real/generated images.
>
> Spatial energy similarity: Gray-level co-occurrence matrix energy.
>
> Spatial complexity and feature magnitude: Derived from UNI2-h embeddings, capturing tissue structure.
>
> Calculated these metrics across P1 C1 and C2 datasets but due to character constraints only presented C1 dataset comparison in the table below. Future work will integrate these metrics to better illustrate and diagnose specific multi-cell model limitations such as boundary definition and texture fidelity.
>
> ### Table: Evaluation Metrics Comparison for C1 Dataset: Multi-cell and Single-cell Models
> || Metric| C1 Multi Diff.| C1 Multi Rect.| C1 Single Diff.| C1 Single Rect.|
> |-----------------------|--------------------------------|---------------------|---------------------|----------------------|----------------------|
> | **Image Quality Metrics ↓** |||| | |
> || FID Overall UNI2-h| 404.06| **87.96**| 405.98 | **39.27**|
> | **Biological Feature Similarity ↑** | ||| | |
> || UNI2-h Embedding Sim.   | 0.967±0.007  | **0.979±0.004**| 0.969±0.007   | **0.983±0.003**|
> || Nuclear Circularity Sim.| 0.835±0.043  | **0.844±0.037**| 0.839±0.049   | **0.874±0.028**|
> || Nuclear Eccentricity Sim.| 0.869±0.046  | **0.954±0.015**| 0.880±0.039   | **0.964±0.012**|
> || Nuclear Solidity Sim.   | 0.714±0.048  | **0.888±0.036**| 0.721±0.077   | **0.867±0.028**|
> | **Spatial Feature Metrics ↑** |||| | |
> || Spatial Energy Sim.| -| **0.283±0.083**| -| **0.678±0.176**|
> || Spatial Complexity Sim. | 0.167±0.138  | **0.506±0.080**| 0.130±0.111   | **0.571±0.085**|
> || Spatial Feat. Magnitude Sim.   | 0.167±0.143  | **0.514±0.077**| 0.135±0.119   | **0.571±0.084**|
>
> > Clearly highlight best results and metric directions?
>
> We appreciate the suggestion to improve table clarity and will highlight all best-performing results in bold in the final version. Additionally, we will include clear metric direction indicators (↑ for higher is better, ↓ for lower is better) alongside each evaluation metric, facilitating intuitive interpretation across tables and speeding reader comprehension.

---

> > ### Comment · Reviewer_Av8N · 2025-08-05
> >
> > Thank you for the author’s responses. I suggest that the paper be revised to include the overall reviewers comments, especially incorporating the updated evaluation metrics which are relevant to the biology domain. In light of these improvements, I have increased my score from 3 to 4.

---

> ### Author Response · Authors · 2025-08-05
>
> Dear Reviewer,
>
> Thank you so much for your thoughtful follow-up and for taking the time to re-evaluate the paper. We're grateful for your reconsideration and for increasing the score! Your constructive suggestions have strengthened our paper.
>
> The final version will incorporate the reviewers' comments and updated evaluations.
>
> Best Regards,
>
> Authors

---

### Official Review · Reviewer_wbMp · 2025-07-05

**Clarity:** 4
**Significance:** 3
**Originality:** 4
**Rating:** 5
**Confidence:** 5

**Summary:**

This paper introduces GeneFlow, a generative model which translates single- or multi-cell spatial trancriptomic profiles into H&E and DAPI histopathology image tiles. The authors frame this as solving the "inverse problem" in computational pathology, which is generating cellular morphology from its molecular state. The model combines an attention-based gene/transcriptomics encoder and a UNet-based rectified flow image decoder. The authors evaluate this model across 3 human melanoma spatial trancriptomics tissues (XeniumC1, C2, and P1). The authors model demonstrate their GeneFlow outperforms a comparable diffusion model in quantitative metrics and a blinded pathologist evaluation. The overarching claim is that this method enables biologically meaningful image synthesis which opens a new direction for genereative models in spatial omics.

**Questions:**

1. Please contextualize the performance gains from using RF adding a comparison to a simpler, deterministic baseline. An example would be something like a conditional UNet using your gene encoder, but then trained to directly regress to the image using MSE or perceptual loss.
2. Please show examples of diverse image generations for the same input transcriptomic vector, sampled with different noise vectors. A good way to quantitatively measure this is through LPIPs or measuring the variance in the perceptual embedding space
3. Please verify through an ablation study the impact of removing components of the encoder. A clear ablation showing performance drops when these components are improved would strengthen the architectural justification, else it may appear overengineered.
4. Please demonstrate examples of failure cases and explain why the multi-cell models underperform the single-cell ones, even though they have more context. Can you show qualitatively how it compares (e.g., does it struggle with cell boundaries, textures, etc.).
5. Please address whether the model can support simple downstream tasks as mentioned in the weaknesses - e.g., recovering cell type annotations from morphology
6. Additional suggestion to show whether model may be causally dependent on certain genes is for a post-hoc extension to select 2-3 top-ranked genes from the attribution study, perturb their input values (e.g., 0, 50, 200% of baseline), and then show change in output morphology or its perceptual embedding.

**Ethical Concerns:**

["NO or VERY MINOR ethics concerns only"]

**Final Justification:**

The rebuttal and discussion period addressed the main concerns from my initial review with meaningful additional experiments, ablations, and biologically relevant evaluation metrics. The inclusion of a deterministic baseline, diversity quantification, and domain-specific measures (such as UNI2-h features, nuclear morphometrics, and cross-modal coherence) significantly strengthens the empirical evidence. The focus on the inverse mapping problem in spatial transcriptomics is novel, well-motivated, and addresses a relevant gap in the field. The improved results demonstrate applicability beyond melanoma, highlighting the method’s generality.

While there is still room to expand the scope of validation and explore explicit spatial coordinate modeling, these are natural directions for follow-up work rather than critical flaws. The manuscript is now clear, technically sound, and positioned to have a meaningful impact on future research in spatial omics. I recommend this work to be accepted.

**Limitations:**

yes

I'd suggest the authors also include a dicussion of:
- How synthetic images might be misinterpreted in downstream pathology workflows
- Whether the model is robust to noise, dropout, or sparsity in input gene profiles

**Quality:**

3

**Strengths And Weaknesses:**

**Strengths**

- The paper is among the first to demonstrate the reverse mapping problem (expression to image) is tractable and biologically meaningful
- The paper distinguishes its goals from RNA-GAN and HistoXGAN (Lines 66-71) which operate on bulk RNA-seq or latent features, avoiding overclaiming and carving a clear case for novelty at the single-cell resolution
- The paper thoughtfully integrates biological context in the encoder architecture (e.g., gene-gene low-rank factorization, global gene attention for long-range, embedding aggregation from neighbouring cells)
- The paper justifies its use of Rectified Flow as an alternative to diffusion
- The paper comprehensively evaluates on on quantitative metrics (FID, SSIM) vs. diffusion baseline
- The paper includes diverse evaluation metrics, including a human expert study (Section 4.4)
- The paper thoughtfully incorporates model interpretability for pathway-level interpretation (Section 4.3) using gradient-based saliency w.r.t. EMT, ECM, which are classic hallmarks and are examples of morphological variation that would manifest in histology
- Figures and overall paper presentation are clear and informative

**Weaknesses**

- Authors describe encoder includes three specialized modules, but no ablation is presented and it is unclear how much each part contributes to performance or whether simpler encoders would suffice
- Line 79 The authors state that rectified flow allows diverse generations from the same transcriptomic input but this is not demonstrated
- Tables 2,3 & Lines 239-254 Authors show the multi-cell models underperform the single cell ones, but doesn't clarify why, nor does it show examples were the model fails. As a reader, I would want to know when not to trust this system (e.g., artifacts, biologically implausible structures)
- The paper lacks comparison to simpler, non-probablistic alternatives and only uses diffusion as a baseline
- The paper gestures at future diagnostic applications, but does not attempt any downstream use case
- Lines 261,262 The paper provides a weak argument for generalizability, claiming the model learns "dataset-agnostic gene-morphology mappings" but limits to three datasets of the same tissue type (melanoma) and technology platform (Xenium)

---

> ### Author Rebuttal · Authors · 2025-07-30
>
> We thank the reviewer for their valuable feedback and high score. We addressed all the raised questions, and performed all requested additional experiments. We have provided extended results for the C1 and new HEST-1k datasets below. Results for P1 and C2 will be available in the final version or provided during the discussion period. Please let us know if further explanation or experiments needed to improve the quality of the final version of the paper.
>
> > Multi-cell models worse than single-cell models? Model produces unrealistic outputs?
>
> Better single-cell model performance (Tables 2–3) from both methodological and biological factors. Unlike multi-cell model using a sliding window, single-cell data is generated from patches centered on valid cells with transcript overlap above a threshold. We apply spatial weighting scheme which gradually decreases pixel weights from center toward margins, emphasizing target cell’s morphology while reducing peripheral influence. Patch sizes cover full cell areas accommodating size variability, yielding more homogeneous, pure-cell patches than multi-cell method. Melanoma-dominated samples compound this imbalance. Single-cell conditioning provides a focused gene expr context. Multi-cell patches include heterogeneous cell types, complicating gene-to-morphology mapping. In melanoma the diverse tumor immune and stromal microenvironment challenges the multi-cell model in resolving cell boundaries and texture consistency. We will include qualitative failure cases in the final version to illustrate these limitations.
>
> We further explored emerging histopathology-specific evaluation metrics using UNI2-h foundational model:
>
> UNI2-h FID: Domain-specific image quality assessment using a pathology foundation model.
>
> UNI2-h embedding similarity: Comparing feature distributions of real and generated images.
>
> Nuclear morphometric similarity: Quantification of circularity, eccentricity, and solidity from segmented nuclei, statistically compared between real/generated images.
>
> Spatial energy similarity: Gray-level co-occurrence matrix energy.
>
> Spatial complexity and feature magnitude: Derived from UNI2-h embeddings, capturing tissue structure.
>
> Calculated these metrics across P1 C1 and C2 datasets but due to character constraints only presented C1 dataset comparison in the table below. Future work will integrate these metrics to better illustrate and diagnose specific multi-cell model limitations such as boundary definition and texture fidelity.
>
> ### Table: Evaluation Metrics Comparison for C1 Dataset: Multi-cell and Single-cell Models
> || Metric| C1 Multi Diff.| C1 Multi Rect.| C1 Single Diff.| C1 Single Rect.|
> |-|-|-|-|-|-|
> | **Image Quality Metrics ↓** ||||||
> || FID Overall UNI2-h| 404.06| **87.96**| 405.98 | **39.27**|
> | **Biological Feature Similarity ↑** ||||||
> || UNI2-h Embedding Sim.   | 0.967±0.007  | **0.979±0.004**| 0.969±0.007   | **0.983±0.003**|
> || Nuclear Circularity Sim.| 0.835±0.043  | **0.844±0.037**| 0.839±0.049   | **0.874±0.028**|
> || Nuclear Eccentricity Sim.| 0.869±0.046  | **0.954±0.015**| 0.880±0.039   | **0.964±0.012**|
> || Nuclear Solidity Sim.   | 0.714±0.048  | **0.888±0.036**| 0.721±0.077   | **0.867±0.028**|
> | **Spatial Feature Metrics ↑** ||||||
> || Spatial Energy Sim.| -| **0.283±0.083**| -| **0.678±0.176**|
> || Spatial Complexity Sim. | 0.167±0.138  | **0.506±0.080**| 0.130±0.111   | **0.571±0.085**|
> || Spatial Feat. Magnitude Sim.   | 0.167±0.143  | **0.514±0.077**| 0.135±0.119   | **0.571±0.084**|
>
> > Comparisons to conditional UNet and ablation on encoder?
>
> Diffusion models are appropriate generative baseline given the many-to-one gene-to-morphology relationship intrinsic to our task; deterministic regression models fail to capture this complexity. Rectified flow offers a simpler, faster alternative to traditional diffusion and regression methods. Following reviewer’s suggestion, we implemented a conditional UNet baseline using our gene encoder trained with MSE and perceptual losses; results in Table below.
>
> Conducted an ablation study comparing transformer based RNA encoder with a simpler encoder in (Sec C2 Fig 7). Our ablation shows the transformer based RNA encoder consistently outperformed the simple encoder in all metrics. Regarding the individual components of our encoder, each of them is uniquely important for the interpretability analysis that we conduct in (Sec 4.3 Fig 4). Removing any component would limit the interpretable aspect of our encoder to identify the most important genes and their relationships. It is seen in the table below that the complexity of the RNA encoder modules is well justified as it takes a small hit in performance but allows for greater interpretability to analyse important gene relations.
>
> Extended experiments to 59 human Xenium samples from 12 organs in the HEST-1k dataset, totaling 1.6M paired patches. Single-cell model is still training, multi-cell evaluations show robust generalization. Cross-dataset results (Tab 4) confirm that training on melanoma alone suffices for good performance despite limited gene overlap, indicating the model captures general gene-morphology mappings rather than dataset-specific artifacts. Figures 8–10 visually corroborate these findings.
>
> ### Table: C1 Single-Cell Ablation Studies and Baseline | HEST-1k Dataset Multi-Cell
> || Metric| UNet (MSE)| Diffusion | Rectified | -Gene Att.| -Gene Att./Rel.| -Gene Att./Multi Att. | HEST-1k|
> |-|-|-|-|-|-|-|-|-|
> | **Image Quality Metrics ↓** || |||||||
> || FID Overall UNI2-h| 525.51| 405.98| 39.27| **26.46** | 27.25| 29.50| 62.54|
> || Inception Feat. Dist.| 21.76±1.29| 20.18±1.61| **14.50±2.52**| 14.89±2.52| 15.02±2.70 | 15.39±2.68| 15.26±2.70 |
> | **Biological Feature Similarity ↑** || |||||||
> || UNI2-h Embedding Sim.| 0.964±0.004 | 0.969±0.007| 0.983±0.003| **0.991±0.003**| 0.990±0.003| 0.990±0.003| 0.974±0.004|
> || Nuclear Circularity Sim.| 0.659±0.038 | 0.839±0.049| 0.874±0.028| **0.949±0.022**| 0.941±0.021| 0.947±0.020| 0.924±0.027|
> || Nuclear Eccentricity Sim.| 0.655±0.024 | 0.880±0.039| 0.964±0.012| **0.959±0.013**| 0.954±0.015| 0.957±0.017| 0.955±0.014|
> || Nuclear Solidity Sim.| 0.479±0.037 | 0.721±0.077| 0.867±0.028| **0.950±0.020**| 0.943±0.019| 0.949±0.020| 0.921±0.028|
> | **Spatial Feature Metrics ↑** || |||||||
> || Spatial Energy Sim.| 0.018±0.034 | - | 0.678±0.176| **0.723±0.130**| 0.735±0.055| 0.719±0.048| 0.744±0.049|
> || Spatial Complexity Sim. | 0.099±0.060 | 0.130±0.111| 0.571±0.085| **0.749±0.069**| 0.704±0.072| 0.721±0.067| 0.641±0.061|
> || Spatial Feat. Magnitude Sim.| 0.112±0.061 | 0.135±0.119| 0.571±0.084| **0.759±0.068**| 0.718±0.068| 0.735±0.069| 0.635±0.059|
>
> > Downstream tasks cell types from morphology?
>
> Collaborating with pathology partners to validate downstream applications, although these are beyond the current study’s scope centered on inverse mapping. Preliminary feedback confirms that pathologists recognize the high fidelity of generated images on melanoma datasets, successfully identifying tumor tissues, cancer-associated fibroblasts, vasculature, and immune infiltration comparable to ground truth. To probe downstream utility, apart from extra evaluations included in the above table, we performed unsupervised clustering using UNI2-h embeddings from generated images, yielding results highly consistent with clustering on real images. This supports that generated images preserve cell composition faithfully. Additionally, our gene importance analysis (Sec 4.3, Fig 4) highlights clinical pathways such as EMT, KRAS signaling, and angiogenesis (App D.1), suggesting the model can prioritize relevant biomarkers and therapeutic targets. The model can be used to predict expected tissue morphology from ST data, enabling consistency checks; deviations between predicted and true histology may flag technical artifacts, tissue quality issues, or biologically meaningful anomalies.
>
> > Diverse image outputs from the same gene input?
>
> Evaluated diversity by generating multiple images conditioned on the same transcriptomic input (dataset C1). Mean LPIPS score 0.388, within the desirable range (0.2–0.5) balancing meaningful variation and biological plausibility. Mean SSIM variance 0.000263, mean PSNR variance 0.372, indicates maintained structural consistency, suggesting the model explores realistic morphological variability rather than producing random artifacts. Quantitative metrics confirm our model’s capability to generate biologically coherent diverse morphologies from identical molecular profiles. Representative diverse images will be included in the final version.
>
> > Changes in gene expr causally affect the generated image?
>
> Our model can perform post-hoc generation under gene expr perturbations. Verification requires controlled perturbation ST datasets currently unavailable. Top 2–3 influential genes may not reflect a complete causal factor, however, causal disentanglement can be integrated in our framework. With the extensive HEST-1k dataset (~60 Xenium samples), future work could compare tumorous versus healthy tissues from the same organ to select candidate genes and verify causal effects. Image-level results of such controlled perturbations will be included in the final version.
>
> > Synthetic images be misinterpreted in pathology?
>
> Xenium spatial transcriptomics data inherently exhibit sparsity and noise, measuring only a subset of genes defined by targeted panels—far fewer than scRNA-seq or bulk RNA-seq. Despite these challenges, our model robustly generates high-quality histological images, demonstrating resilience to gene expression bias, noise, dropout, and input sparsity. Notably, we trained on the C1/C2 datasets (~300 genes) and evaluated on the P1 dataset, which includes ~5,000 genes with only 126 overlapping, and observed strong generalization and performance (Table 4, Supplementary Figures 9 & 10). These results show the model’s practical applicability and robustness for real-world spatial transcriptomics workflows.

---

> ### Comment · Reviewer_wbMp · 2025-08-05
>
> Thank you for your rebuttal and the additional experiments provided. The ablation study, the new deterministic baseline, the diversity quantification (LPIPS), and the deeper feature analysis have addressed my main technical concerns about the architectural justification and the rigour of the evaluation.
>
> I understand the limitations of the gene perturbation study's data and the reasons behind the multi-cell model's performance. As noted in the rebuttal, these analyses have been deferred.
>
> The authors have sufficiently addressed the critical points from my review. The paper is strengthened by these additions. I will maintain my original rating.

---

> > ### Author Response · Authors · 2025-08-05
> >
> > Dear Reviewer,
> >
> > Thank you for your follow-up and for acknowledging the additional analyses and clarifications we provided.
> >
> > We appreciate your recognition of the new experiments and the improvements made to the paper based on your feedback. We're glad that the updates addressed your main concerns and helped strengthen the overall contribution.
> >
> > Best Regards,
> >
> > Authors

---

### Note · Authors · 2025-08-12

We sincerely thank all reviewers for their thoughtful engagement and constructive feedback, which helped clarify our contributions, strengthen our evidence, and refine future directions.

All three recognized major strengths: novel problem formulation for inverse mapping from gene expression to histology (Uins), the first tractable and biologically meaningful reverse mapping (wbMp), and methodological innovation in applying generative models to spatial biology (Av8N). Two reviewers raised their scores after the rebuttal (Av8N: 3 to 4, Uins: 4 to 5), while wbMp maintained a strong Accept (5), reflecting the positive feedback of the work.

In the rebuttal, we demonstrated the necessity of our rectified flow architecture through further ablations against diffusion and conditional UNet baselines, achieving 3-6× lower FID. We addressed evaluation concerns by introducing domain-specific metrics via the UNI2-h foundation model and nuclear morphometrics.

Robust generalization was shown on the extended HEST-1k dataset (yielding 1.6M patches from 59 samples across 12 organs, ) despite minimal gene overlap (~126/5000), indicating dataset-agnostic gene–morphology mappings. Biological credibility was confirmed through 86\% expert pathologist preference in blinded comparisons, high cross-modal coherence with HE2RNA (r > 0.98), and identification of clinically relevant pathways (EMT, KRAS signaling, angiogenesis) via gene importance analysis.

Reviewer feedback also prompted us to strengthen the spatial awareness of our multi-cell mode. While our design already incorporates a cell-boundary masking channel alongside RGB and auxiliary inputs, it was excluded from main results due to segmentation limitations in 10x Xenium preprocessing. Building on these suggestions, we plan to construct cell-graph representations from mask centroids and apply invariant/equivariant GNNs to introduce explicit spatial constraints. We are currently implementing it and will include the evaluation in the final version.

Overall, GeneFlow is the first comprehensive framework for inverse mapping in spatial transcriptomics, capturing fundamental gene-morphology relationships and enabling predictive visualization, biomarker discovery, and improved quality control. We are grateful for the reviewers' recognition of its novelty, rigor, and potential impact across AI and computational biology.

---

### Decision · Program_Chairs · 2025-09-17

**Decision:**

Accept (poster)

**Comment:**

The paper proposes an end-to-end framework for generating histopathological images directly from single- or multi-cell gene expression data. The method combines an attention-based RNA encoder with a conditional UNet guided by rectified flow dynamics. The method outperforms diffusion and deterministic baselines across FID, SSIM, UNI2-h embeddings, nuclear morphometrics, and in pathologist evaluations (several metrics added in the rebuttal phase).

The consensus among reviewers is that the paper makes a novel and relevant contribution by tackling the inverse mapping from gene expression to histology, introducing (well-known) rectified flow modeling with bespoke network components. It is supported by strong empirical results across datasets and evaluation metrics (some added in the rebuttal phase), including pathologist validation and pathway-level interpretability. The method generalizes well to diverse tissues, and the work is clearly presented and accessible.

The main weaknesses include the lack of explicit spatial coordinate modelling, which limits biological plausibility, and ambiguity around the biological meaning of single-cell image generation. The original submission relied heavily on generic metrics and lacked simple baselines or ablations, though these were later added. Analysis of failure cases, particularly for multi-cell mode, is limited, and downstream diagnostic applications remain largely speculative. The weaknesses (lack of explicit spatial modeling, causal validation, and limited downstream applications) are acknowledged but framed as natural directions for future work rather than critical flaws. The authors have promised to include an additional study on the spatial aspects of the multi-cell mode.

Overall, the reviewers agree that the paper contributes a novel approach to spatial transcriptomics, and the focus on the inverse mapping problem in spatial transcriptomics is novel, well-motivated, and addresses a relevant gap in the field. It provides rigorous methodological and empirical advances and demonstrates strong biological validation. While there are several updates required (and promised) for the camera-ready version of the paper, the paper’s combination of novelty, technical soundness, and potential impact makes it relevant to the machine learning for science community at NeurIPS.

I recommend acceptance.